# Flexibility in motor timing constrains the topology and dynamics of pattern generator circuits

Cengiz Pehlevan[1], Farhan Ali[2,3,4] & Bence P. Ölveczky[2,3]

Temporally precise movement patterns underlie many motor skills and innate actions, yet the flexibility with which the timing of such stereotyped behaviors can be modified is poorly understood. To probe this, we induce adaptive changes to the temporal structure of birdsong. We find that the duration of specific song segments can be modified without affecting the timing in other parts of the song. We derive formal prescriptions for how neural networks can implement such flexible motor timing. We find that randomly connected recurrent networks, a common approximation for how neocortex is wired, do not generally conform to these, though certain implementations can approximate them. We show that feedforward networks, by virtue of their one-to-one mapping between network activity and time, are better suited. Our study provides general prescriptions for pattern generator networks that implement flexible motor timing, an important aspect of many motor skills, including birdsong and human speech.

[1] Center for Computational Biology, Flatiron Institute, New York, NY 10010, USA. [2] Department of Organismic and Evolutionary Biology, Harvard University, Cambridge, MA 02138, USA. [3] Center for Brain Science, Harvard University, Cambridge, MA 02138, USA. [4] Present address: Department of Psychiatry, Yale University School of Medicine, New Haven, CT 06511, USA. These authors contributed equally: Cengiz Pehlevan, Farhan Ali. Correspondence and requests for materials should be addressed to C.P. (email: cpehlevan@flatironinstitute.org)

The brain's ability to generate spatiotemporally precise motor patterns underlies much of what we do, from the expert performances of athletes and musicians to the daily acts of speaking or walking. Despite the ubiquity and importance of stereotyped motor sequences, our understanding of how the brain controls and modifies their timing remains poor[1,2].

Many different neural network architectures can generate temporally reproducible dynamics, a minimum requirement for producing precisely timed motor output[3–9]. Randomly connected recurrent neural networks (RNNs), a common approximation for how neocortex is wired[10], can be trained to generate prescribed temporal patterns[7,9,11]. Feedforward networks, such as synfire chains[3,4], naturally map network activity to elapsed time, and hence are also suited for generating timing signals underlying stereotyped motor sequences[1].

However, the demands on pattern generator networks often go beyond simply producing the same dynamics every time an action is executed. A characteristic of skilled performance is the flexibility with which established motor patterns can be modified, including changes to their timing[12–14]. Modifying the overall tempo of a continuous action sequence is an obvious and often discussed example[15,16] (Fig. 1a). However, careful analysis of stereotyped action sequences suggests that the nervous system may be capable of altering their timing in far more specific and flexible ways (Fig. 1a). In speech production or typing, for example, the duration of discrete segments (e.g. phonemes or elements of key strokes) can change independently of other parts of a continuous action sequence[14,17]. Such flexibility in motor timing can be essential for improving motor skills and adapting them to new contingencies or task demands. Increasing the power of your tennis serve, for example, may require selectively speeding up the power-generating stroke, while leaving the timing of the rest of the serve unchanged. Learning the speech patterns of a new language can similarly benefit from the ability to alter the duration of distinct phonemes and syllables in context-specific and flexible ways[18].

To better characterize the flexibility with which the brain can modify the timing of stereotyped motor sequences, we took advantage of an experimental paradigm that allows adaptive changes to the temporal structure of zebra finch song to be induced[19]. The complex learned vocalizations of adult songbirds are in many ways similar to human speech[20,21], being temporally precise and under certain experimental conditions also quite malleable[19,22,23]. Moreover, the temporal precision and reproducibility of birdsong are of ethological importance[24] as is the capacity to change the song's temporal structure[12]. These attributes make the adult zebra finch a good model for interrogating the flexibility with which the timing of complex and reproducible motor sequences can be modified.

Here we show that changes to the duration of specific song segments do not interfere with the temporal structure of other segments, and that two segments in the same song can be modified simultaneously and independently. Such flexibility in motor timing constrains the functional architecture of the underlying pattern generator circuits. We derive formal prescriptions for how such specific and adaptive changes to motor timing can be implemented in neural circuits. We show that for the duration of two intervals in a motor sequence to change independently, their respective 'gradient' vectors in synaptic weight space must be orthogonal to each other (a gradient vector's direction is the direction of maximal duration increase while its magnitude represents how fast the duration increases in that direction). We probe the extent to which neural networks capable of generating precise temporal patterns conform to this prescription. While RNNs, in general, do not fulfill the strict conditions of orthogonality, certain implementations can achieve

reasonable approximations. However, we find that feedforward networks, which unambiguously map network activity to time, are better suited to implement flexible motor timing.

Consistent with our theoretical analysis, simulations of temporal learning using biologically plausible plasticity rules in a synfire chain network reproduced the phenomenology observed in songbirds, while similar simulations in RNNs failed to attain the required degree of temporal flexibility. The different network simulations also generated predictions for how variability in the duration of a song segment should change with modifications to its mean duration. Here too, our experimental data conformed to the predictions of the synfire chain model, but not those of the RNNs. These results constrain the topology and dynamics of neural networks underlying flexible motor timing, and suggest that the synfire chain architecture may be ideally suited for its implementation. Even though the experimental data come from songbirds, we note that the theoretical constraints on temporally flexible pattern generator circuits we derive are very general and apply also to other systems.

## Results

**Flexible modifications to the temporal structure of birdsong.** Flexibility in motor timing can be characterized in terms of specificity and independence. Specificity means that changes to the duration of one part of a sequence leaves the temporal structure of other parts unaffected (Fig. 1a). Independence means that the timing in different parts of a sequence can be modified simultaneously without any effect on learning rates. We studied both aspects of timing flexibility in perhaps the best understood example of how the brain acquires and executes complex learned motor sequences: the courtship song of the zebra finch[25,26].

To characterize the flexibility with which the pattern generator circuit underlying stereotyped adult song can modify motor timing (Fig. 1a), we challenged adult birds to produce changes to the duration of specific song segments (Fig. 1b), using Conditional Auditory Feedback (CAF), a reinforcement learning-based paradigm[19,27] (Methods). By playing loud (~80−90 dB) aversive sound bursts contingent on the duration of a targeted song segment, significant changes to the song's temporal structure can be induced in a matter of hours[19] (Fig. 1c). To explore the specificity of these modifications, we compared the changes in the duration of targeted segments to the changes in other parts of the song.

Across the population of birds ($n = 18$ birds), targeted segments changed by, on average, $2.9 \pm 1.7$ ms (mean ± SD, $n = 18$ targets) per day relative to baseline drift (Fig. 1d, Methods). In contrast, we saw no significant changes in non-targeted segments ($n = 120$) regardless of their position relative to the targeted segment (Fig. 1d, Supplementary Fig. 1A). Previous analyses have shown correlations among song segment durations during baseline undirected singing[28,29]. However, more detailed analysis found that specificity in the CAF-induced timing changes was not related to baseline timing correlations (Supplementary Fig. 1B, 1C). This demonstrates that the timing of stereotyped learned motor patterns can change in temporally very specific ways.

We next tested whether learning-induced changes to the duration of one song segment interferes with adaptive modifications to other parts of the song by targeting two segments (>100 ms apart) for CAF simultaneously (Fig. 1e, Methods). Interestingly, we found no difference in learning rates whether a segment was targeted for modification alone ($2.8 \pm 2.4$ ms per day) or in conjunction with another segment ($3.0 \pm 2.4$ ms per day, $n = 12$ segments in 6 birds, Fig. 1f, g), suggesting that individual segments of song are, in terms of changes to their durations, independent.

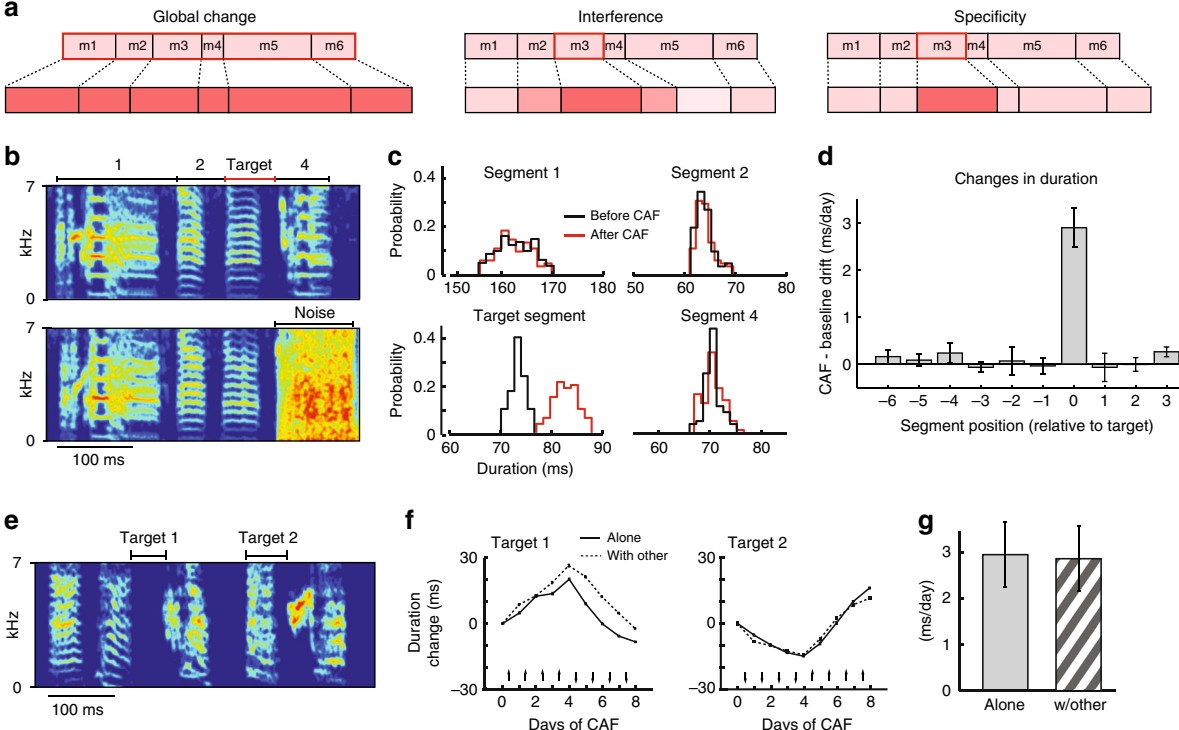

**Fig. 1** Flexible modification of motor timing in songbirds. **a** Schematic showing different ways in which the temporal structure of a motor sequence (encompassing motor segments m1–m6) can change. (Left) Global change—duration of all segments change together. (Middle) Interference—modifications to the duration of one segment (m3, red box) interferes with (i.e., changes) the duration of other segments. (Right) Specificity—each segment can change independently of others. **b** (Top) Spectrogram of a zebra finch song, divided into four segments. One of these ('target') was marked for lengthening/shortening by delivering aversive noise bursts after the end of the target segment when its duration was below/above a threshold (Bottom, see Methods). **c** Duration distributions for the segments in **b** when the target segment was lengthened by our CAF paradigm. **d** Summary statistics (mean ± SEM) of the daily changes to the duration of target and non-target segments ($N = 18$ targeted segments and $n = 120$ non-targeted segments in 18 birds). Changes in target intervals differed significantly from zero ($p = 1.6 \times 10^{-6}$, one-sample t-test), whereas non-targets did not ($0.24 \leq p \leq 0.98$, one-sample t-test). We used the same number of days of baseline (before CAF was started) as the number of days of CAF, which for each bird varied between 6 and 10 days. **e** Spectrogram of a song for which two segments were targeted for modification, either alone or in conjunction with each other. **f** Learning trajectories (mean ± SEM) from an example bird during CAF experiments designed to change the duration of the targets in **e**, either alone or together. Arrows denote the direction in which the duration of the targets were induced to change by the CAF paradigm. **g** Summary statistics (mean ± SEM) from our population of birds of the daily changes to the duration of targeted segments, as a function of whether they were targeted alone or together with another segment ($n = 12$ segments in 6 birds, $p = 0.54$, paired-samples t-test; see Methods)

**Constraints imposed on flexible time-keeper circuits.** While many different neural network architectures have been proposed to generate reproducible temporal patterns[1,7,9], the extent to which they support flexibility in motor timing has not been considered.

Here we use theoretical and computational approaches to examine how specificity and independence in motor timing constrain the topology and dynamics of the underlying pattern generator networks. We assume that network output is modified by changing connection strengths between its neurons, and seek to understand the network properties that enable flexible motor timing.

To modify the duration of a discrete segment in a longer action sequence with independence and specificity (Fig. 1a), there must exist a path in synaptic weight space along which only the duration of that segment (i.e., its interval) changes (specificity). Moreover, changes along such a path should not affect the network's capacity to change along paths associated with other intervals (independence). The analytical discussion that this condition can be satisfied in a generic pattern generator network is presented in Supplementary Note 1.

However, the existence of specific and independent paths in synaptic weight space is, on its own, not sufficient to ensure

flexible motor timing. Such paths must be found and followed by biologically plausible learning algorithms. For real-world pattern generators, exemplified by the vocal control pathway of the zebra finch, modifications to motor timing often happens through trial-and-error learning, i.e., variations in temporal structure are generated and certain variants reinforced[19]. Our experiments with zebra finches showed that reinforcement targeted to specific song segments produce specific and independent changes to the song's temporal structure (Fig. 1).

Synaptic learning rules that implement reinforcement learning typically find the reward gradient vector, i.e., the vector in synaptic weight space that points along the direction that maximizes reward increase (or punishment decrease), and updates synaptic weights along that direction[30–33]. To implement such trial-and-error learning, network dynamics must vary from rendition-to-rendition, with dynamics resulting in higher rewards being reinforced by changing synaptic weights appropriately.

Important to note is that such learning rules will only find the exact reward gradient after averaging synaptic weight updates over many trials, a luxury not afforded a learning animal. In reality, the synaptic updates are noisy approximations of the true gradient. In the next sections, we will explicitly simulate trial-and-error learning in the temporal domain, comparing

implementations in different network architectures. Here, we focus on the gradient directions themselves, assuming that they can be found, and probe if their geometries allow for flexible time keeping. This allows us to separate two factors that influence and determine timing flexibility: (1) the geometry of gradient vectors, purely a property of the network, and (2) the constraints of working with noisy approximations of the gradient, which arises due to trial-and-error learning.

Let $R^{(\alpha)}(I^{(\alpha)})$ denote a scalar reward signal associated with the duration of an interval $\alpha$, which we denote by $I^{(\alpha)}$. To state the condition for specificity in a pattern generator network, we make use of two facts. First, the reward gradient is parallel to the interval duration gradient (a fact that follows from the chain rule of calculus). Second, while the gradient points along the direction of maximal increase in interval duration, directions perpendicular to it are directions along which the duration of the interval does not change. Then, specificity requires an interval gradient to be orthogonal to all other interval gradients. Formally:

$$\sum_{\{ij\}} \frac{\partial I^{(\alpha)}}{\partial W_{ij}} \frac{\partial I^{(\beta)}}{\partial W_{ij}} = 0, \text{ if } \alpha \neq \beta, \qquad (1)$$

where $W_{ij}$ is the strength of the connection from neuron $j$ to neuron $i$, and the summation being over all existing synapses (see also Methods and Supplementary Note 1). Importantly, the orthogonality constraint must hold throughout learning. Otherwise, a network that initially exhibits specificity could lose that property as the target segment (and the gradient vector associated with it) changes. See Methods and Supplementary Note 1 for a comprehensive discussion of these points. Importantly, Eq. (1) also ensures independent learning, assuming that the effects of separately delivered reinforcers sum linearly (see Supplementary Note 1).

**Quantifying timing flexibility**. We next asked whether neural network architectures proposed to underlie the temporal patterning of action sequences conform to the prescriptions we derived. We consider two broad classes of networks: RNNs[7,9,10] and feedforward networks[3,4,34,35], and investigate the condition under which instantiations of these networks satisfy or approximate the orthogonality condition in Eq. (1). To quantify the extent to which a network deviates from the specificity and independence criterion (Eq. 1), we numerically calculate the gradient vector associated with each interval and then compute the "interference matrix". This is a symmetric matrix whose elements are given by the inner products of the gradients associated with pairs of targeted segments:

$$M_{\alpha\beta} = \sum_{\{ij\}} \frac{\partial I^{(\alpha)}}{\partial W_{ij}} \frac{\partial I^{(\beta)}}{\partial W_{ij}}. \qquad (2)$$

This matrix has non-zero off-diagonal elements when gradients are non-orthogonal.

The interference matrix of a pattern generator network quantifies the degree to which the network is flexible. For example, consider an 'experiment' akin to the one we performed in songbirds (Fig. 1), where the duration of a segment $\alpha$ (i.e., its 'interval') is targeted for modification. For small changes in synaptic weights, the expected change to the duration of another segment $\beta$, normalized by the change in the target interval $\alpha$, is given by:

$$\frac{\delta I^{(\beta)}}{\delta I^{(\alpha)}} = \frac{M_{\beta\alpha}}{M_{\alpha\alpha}}. \qquad (3)$$

Eq. (3) shows how the interference matrix relates to specificity. If

the gradients with respect to intervals $\alpha$ and $\beta$ are non-orthogonal, then $M_{\beta\alpha}$ is non-zero and there is a change in interval $\beta$, even though the reward was targeted at $\alpha$. In these cases, timing modifications will not be specific to the targeted interval. In an 'experiment' where two intervals ($\alpha$ and $\beta$) are targeted for modification simultaneously with separately delivered reinforcers, the expected change in interval $\alpha$, normalized to the expected change in an 'experiment' where only interval $\alpha$ is targeted, is given by:

$$\frac{\delta I^{(\alpha)}_{\text{two-tar.}}}{\delta I^{(\alpha)}_{\text{single-tar.}}} = 1 + \frac{M_{\beta\alpha}}{M_{\alpha\alpha}} \frac{\frac{\mathrm{d}R^{\beta}}{\mathrm{d}I^{\beta}}}{\frac{\mathrm{d}R^{\alpha}}{\mathrm{d}I^{\alpha}}}. \qquad (4)$$

Eq. (4) shows how the interference matrix relates to independence. If the gradients with respect to intervals $\alpha$ and $\beta$ are non-orthogonal, then $M_{\beta\alpha}$ is non-zero, meaning that learning directed at interval $\beta$ will affect the learning rate of interval $\alpha$. In these instances, the temporal modifications to different segments will not be independent. We note that the ratio $M_{\beta\alpha}/M_{\alpha\alpha}$ appears in both Eqs. (3) and (4) and quantifies the deviation from specificity and independence. Below, we use it, or rather its absolute value, as a measure of interference (or non-flexibility) between intervals $\alpha$ and $\beta$. Therefore, with the knowledge of the interference matrix, we can quantify the degree to which a time-keeper network allows for flexible motor timing.

**Probing random recurrent networks as flexible time-keepers**. RNNs in their chaotic regime[10] produce spatiotemporal patterns suitable for generating complex motor output. However, chaotic network dynamics cannot produce the same dynamics repeatedly and reliably because even small perturbations to the network can cause large changes in its dynamics. Recently, algorithms that suppress chaos in RNNs[7,9] have demonstrated the capacity of such networks to learn and reliably produce complex stereotyped output, making them suitable as pattern generators. Next, we examine the extent to which such networks, specifically the feedback-stabilized RNNs of Sussillo and Abbott[9] and the dynamic attractor networks of Laje and Bounomano[7], support flexibility in motor timing.

**Robustness–flexibility trade-off in feedback-stabilized RNNs**. Using the FORCE algorithm[9], we trained the read-out units of an RNN with 500 'rate' neurons to produce an output pattern with readily identifiable 'intervals' (Fig. 2a). In this network, chaos is suppressed by having the linear read-out neuron feed back to neurons in the RNN[9,36], and hence we refer to it as "feedback-stabilized RNN" or fsRNN. For simplicity, we trained the network to produce a waveform with 10 equally spaced peaks, 50 ms apart (Fig. 2a, Supplementary Fig. 2A, Methods). This allowed us to define the start and end of specific 'intervals' as the times the output crossed a preset threshold.

In fsRNNs, feedback from the read-out unit to the neurons in the RNN should be comparable in strength to the recurrent inputs to those neurons in order to suppress chaos and allow for successful training[9]. In our implementation, the strength of the feedback is governed by the parameter $g_{\text{FB}}$, which is typically set to be around 1 in applications of FORCE training RNNs[9,37–39]. We varied this number from 0 to 5 and trained 20 networks for each value. We found that increased feedback improves training success as reflected by a decrease in the 'test' error, i.e., the normalized error between network output and desired output (Fig. 2b, Methods). To assess whether such networks can produce precise timing signals, we quantified the fraction of simulations (out of 400) in which the network failed to produce all 10 intervals within 3 ms (i.e., 6%) of the target duration (the 'timing

failure rate', Fig. 2b). This metric was close to zero even when the strength of the feedback was low, meaning that even networks with outputs further from the specified target can produce activity modulations useful for marking the starts and ends of intervals.

Looking at the networks more closely, we found that changes in feedback strength led to qualitatively different solutions (Fig. 2a). When feedback was strong, it dominated the recurrent input to the pattern generator network close to interval boundaries. Because the feedback input was periodic, the network state, defined by the

vector of all instantaneous unit activities, was reset to the same state at interval boundaries (Fig. 2c), leading to periodic activity in the pattern generator network (see example unit dynamics in a network with $g_{FB} = 2$, Fig. 2a). When feedback strength was low, the recurrent input dominated, causing non-periodic network activity (see example unit dynamics in a network with $g_{FB} = 1$, Fig. 2a). Given these qualitatively different dynamic regimes, we probed RNNs with varying degrees of feedback in terms of their capacity for robust and flexible time-keeping.

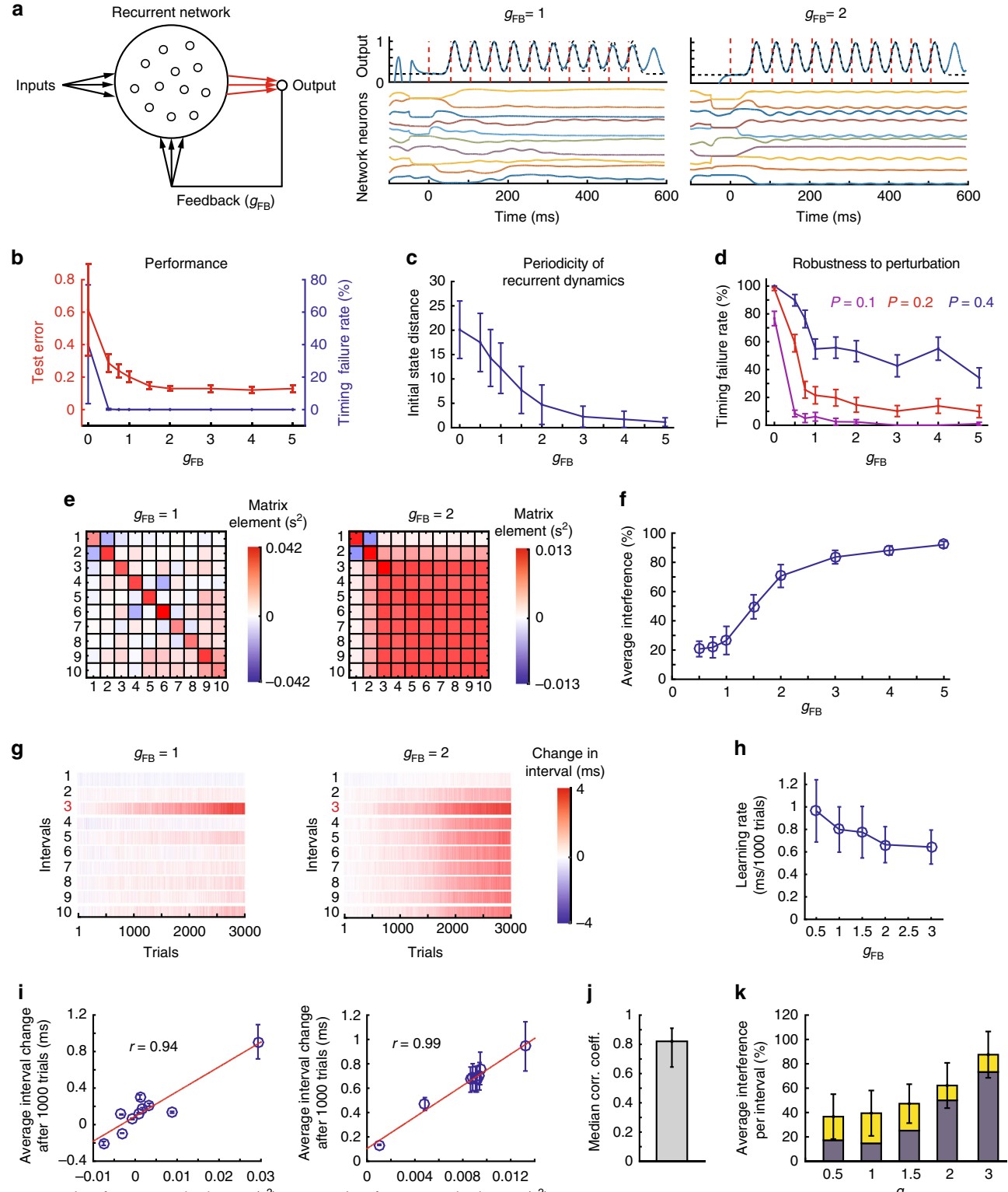

We first quantified the robustness of the network as a function of feedback by injecting 10 ms perturbation pulses of different magnitudes into all units of the network similar to Laje and Buonomano[7] (Fig. 2d, Supplementary Fig. 2B). We applied this perturbation during the second interval and found that the timing failure rate decreased with increased feedback, though not much beyond $g_{FB} = 1$. This implies that networks become more robust as the strength of feedback is increased.

We next examined how these networks cope with demands for timing flexibility by calculating their interference matrices (Fig. 2e). We found that interference matrices are sensitive to the strength of the feedback, $g_{FB}$: at low feedback, off-diagonal elements are small, implying relatively low interference between intervals and high specificity. At high feedback, off-diagonal elements are almost as large as diagonal elements, implying high interference and low specificity. This is because the dynamics of the network as well as the consequences of the synaptic changes are similar for each interval. Interestingly, at high feedback, the first interval behaves differently than the others, and interferes much less. This is because the network state at the beginning of the first interval, set by an external start signal, is different than the network states at the beginning of other intervals (Fig. 2c), which are determined largely by the feedback signal.

We quantified timing flexibility by calculating the normalized interference matrix elements, $\left|\frac{M_{\beta\alpha}}{M_{\alpha\alpha}}\right| \times 100\%$, which reflect interference for all pairs of intervals as discussed above. On average, interference went from ~20% to almost 100% as feedback increased (Fig. 2f). Taken together, our results (Figs. 2c−f) show that feedback leads to a trade-off between robustness and timing flexibility, with strong feedback making the network more robust to perturbations, but less flexible in the time domain.

Thus far, we have inferred flexibility from gradient vectors in synaptic weight space and metrics derived from those. Biological implementations of reinforcement learning, however, must find those gradients. To understand how this affects timing flexibility, we simulated the process of reinforcement learning by implementing a biologically plausible rule that updates synaptic weights to maximize reward on average[31,32] (Methods). Noise added to the network units, both during training and later, ensured trial-by-trial variability in interval durations to the level observed in songbirds[28] (Methods). Reinforcement was provided to the network in a manner analogous to our CAF experiments: if the target interval differed from the running average in the desired direction (Methods), reinforcement was delivered at the end of the interval (Methods). This learning rule successfully modified the duration of targeted intervals, with learning rates comparable to those seen during CAF (Fig. 2g, h). When averaged across multiple runs of the simulated learning 'experiments', the interference patterns

were largely predicted by and consistent with the interference matrix calculations (Fig. 2i, j). However, interference in single 'experiments' exceeded the predictions from gradient descent, i.e., the interference matrix, by a significant amount (Fig. 2k), suggesting that experiment-to-experiment variability in the modification of non-target intervals is large. The single experiment interference decreased with decreasing feedback and leveled around 35% (averaged across intervals, networks, and simulations, Fig. 2k, Methods).

To assess independence in the temporal domain, we targeted two intervals for modification as we had done experimentally in songbirds (Fig. 1e−g), with separate reinforcements given contingent on the duration of both intervals (Methods). As expected from the interference matrices, this resulted in destructive interference, meaning that learning rates for individual intervals decreased when they were modified in conjunction with another (Supplementary Figures 2C−E). This interference increased with feedback, leading to almost no learning for $g_{FB} > 2$ (Supplementary Figures 2C−E).

We next asked whether interference can be prevented by an internally generated "template" reinforcer, which clamps non-target interval durations to their baseline value? Such reinforcer is hinted at by the experiments of Ali et al.[19], where interval durations modified by the CAF procedure returned to baseline values after CAF was turned off. To test this hypothesis, we introduced a separate reinforcer for each interval, which rewarded interval duration changes toward the interval's baseline value. Such reinforcer successfully restored the target interval's baseline duration after the CAF-like process implemented in our simulations was turned off, with the rate of change comparable to what is seen in songbirds[19] (Supplementary Fig. 2F). However, the overall level of interference did not systematically decrease with the inclusion of template reinforcers (Supplementary Fig. 2G), because they themselves interfered with each other and the CAF process.

**Time-keeping with dynamic attractor networks.** Recently, Laje and Buonomano[7] proposed a new dynamic regime for RNNs, the dynamic attractor, which they showed can function as a "population clock" suitable for generating temporally precise output due to its robustness to perturbations. However, the extent to which the dynamic attractor allows flexible motor timing has not been considered.

The dynamic attractor has a similar architecture to the fsRNN, but differs in that it has no feedback from the read-out neuron (Fig. 3a). To ensure robustness, the RNN is instead trained using an "innate training" procedure[7], which applies the FORCE algorithm to neurons in the pattern generator network in order to

**Fig. 2** Feedback-stabilized RNNs exhibit a robustness–flexibility trade-off. **a** (Left) The network architecture. Trained connections are in red. (Middle, right) Firing rates of example neurons and the network outputs. **b** Training performance (test error, Methods) and timing failure rate (the frequency with which the network fails to produce all 10 intervals within 6% of their target durations over 400 trials) versus feedback strength. **c** The distance between two instantaneous population firing rate vectors, each constructed at the beginning of an interval (Methods), averaged over all interval pairs for intervals 3–10, 20 fsRNNs and 100 runs over each fsRNN. **d** Timing failure rate in response to a perturbation pulse ($P$) versus feedback strength. Error bars show standard error across 20 fsRNNs. **e** Example interference matrices for fsRNNS. **f** Average interference calculated from the interference matrix. The plot shows averages over all pairings of intervals 2–10 for 20 fsRNNs at each feedback strength. **g** Examples of a reinforcement learning 'experiment' that targeted interval 3 for lengthening. **h** Learning rate (the absolute value of the change in the target interval duration after 1000 trials) averaged over 20 learning 'experiments' in 20 trained networks. **i** Interval duration changes (Methods) after 1000 trials for the 'experiments' in **g**, averaged across 20 different simulations, regressed to the elements of the 3rd row (i.e., those associated with the target interval) of the networks' interference matrices. Error bars show standard error across 20 simulations; $r$ denotes Pearson correlation coefficient. **j** The analysis of **i** applied to 100 fsRNNs (20 trained networks for each of $g_{FB} = 0.5, 1, 1.5, 2, 3$), two times each, one for stretching and one for shrinking the 3rd interval. The median correlation coefficient over the 200 regressions performed is plotted. Error bars denote 25th and 75th percentiles, respectively. **k** Change in non-targeted intervals in the first 1000 trials of reinforcement learning simulations, relative to the change in the target interval (interference). Average over all non-target intervals and 20 simulations across 20 fsRNNs as a function of feedback strength. The blue portion of the bars show the average interference calculated from the interference matrix (as in **f**)

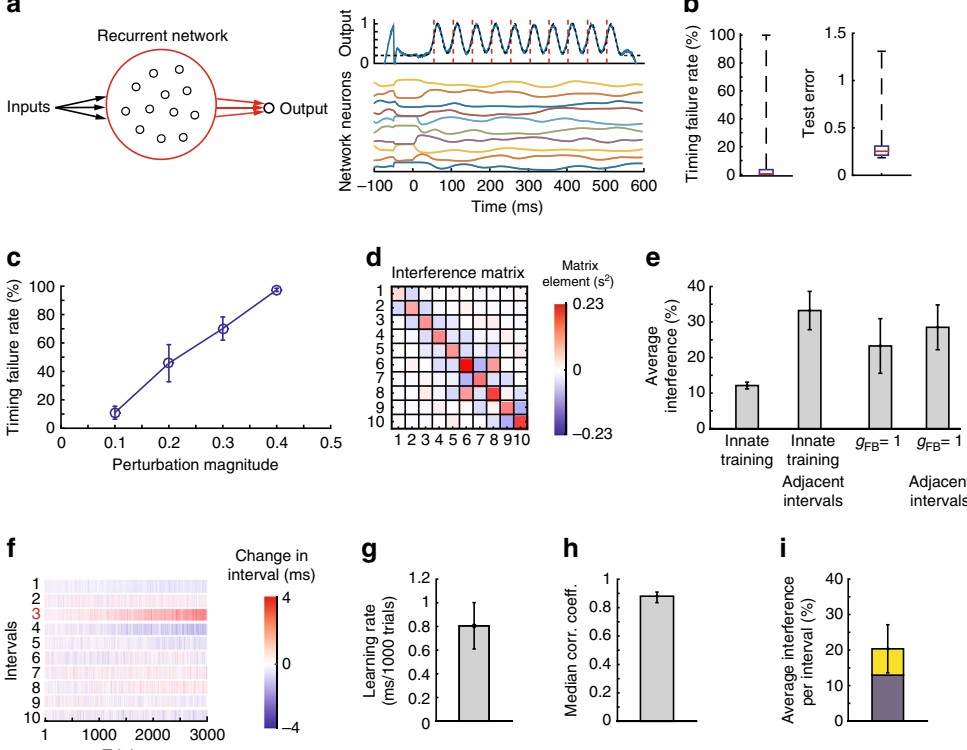

**Fig. 3** Flexible timing with dynamic attractor networks. **a** Schematic showing the network architecture. Recurrent connections as well as the connections to the output neuron are trained. On the right are the activities of example neurons from the pattern generator network and the output neuron. **b** Box plots of training (test error, Methods) and timing (timing failure rate out of 400 trials) performances of 20 trained networks. Whiskers show maxima and minima; 14 out of 20 networks were successfully trained and used for the subsequent analysis in this figure. **c** Timing failure rates (out of 400 trials) for the network when perturbed by a pulse input of varying magnitude. **d** Example interference matrix. **e** Relative change in the duration of non-targeted intervals as a function of the change in targeted intervals ('average interference'), calculated from the off-diagonal elements of the interference matrix, as in Fig. 2f except that the 1st interval is also taken into account. The average interference is compared to the interference in intervals adjacent to the target, and for fsRNNs with $g_{FB} = 1$. **f** Example of a reinforcement learning 'experiment' that targeted interval 3 for lengthening. Shown is the effect on all intervals after for 3000 trials. **g** Learning rate averaged across networks as in Fig. 2i. **h** Median correlation coefficient between the interval changes calculated from the interference matrix and from the reinforcement learning simulations (as in Fig. 2i). Error bars denote 25th and 75th percentiles, respectively. **i** Average interference in reinforcement learning simulations as in Fig. 2k. The blue portion of the bars show the average interference calculated from the interference matrix

stabilize a chaotic trajectory already produced by the network. This effectively turns the trajectory into an attractor such that even if the network is perturbed, it returns to the original trajectory, yielding a reliable and precise output pattern.

We trained instantiations of these dynamic attractor networks, matched in size and other parameters to those of the fsRNNs (Methods), to produce output patterns from which interval durations could readily be extracted (Fig. 3a, Methods). We found that a large fraction of the trained networks successfully produced all 10 intervals within 3 ms of their target duration (Fig. 3b). In our analysis, we only considered networks that were successful >99% of the time (out of 400 simulations, 14 out of 20 trained networks). The robustness of these networks to perturbations was comparable to fsRNNs with low feedback ($g_{FB} \approx 1$), i.e., similar perturbation magnitudes led to similar failure rates (Fig. 3c). These results ensured that our dynamic attractor networks were comparable to fsRNNs in reliability and robustness, allowing us to compare them in terms of their capacity to support flexible motor timing.

Interference matrices revealed less interference across intervals (~12%) than the low feedback RNNs (~23%) (Fig. 3d, e). In contrast to the fsRNNs trained by the FORCE learning algorithm, the interference between adjacent intervals was, on average, much larger (~35%) than the average interference (Fig. 3e). This is

consistent with the dynamic attractor trying to 'return' to its original pattern, compensating for a decrease/increase in the duration of an interval by increasing/decreasing the duration of adjacent intervals. In line with this, the interference matrix elements corresponding to neighboring intervals consistently had negative values (Fig. 3d).

When we modified the duration of targeted intervals with the same reinforcement learning algorithm that we used for the fsRNNs (Methods), learning rates were similar to the simulated experiments in fsRNNs and comparable to what we observed in songbirds (Fig. 3f, g). Changes in the duration of target and non-target intervals were, over all, well predicted by the interference matrices (Fig. 3h), though as with the fsRNNs, interference on average (~20%) exceeded what was expected by gradient descent (~12%) (Fig. 3i).

To assess independence, we again targeted two non-adjacent intervals for simultaneous modification. We found that the learning changed as predicted by the associated interference matrices (Supplementary Fig. 3A–C). Absolute changes in learning rates were less (~15%) than for low feedback fsRNNs (~30% for $g_{FB} = 1$), consistent with dynamic attractors exhibiting less interference on average. Including template reinforcers (Supplementary Fig. 3D, E) did not lead to a systematic reduction in interference.

These results suggest that dynamic attractor networks, as in Laje and Buonomano[7], allow for more flexible adjustments to motor timing than fsRNNs. In contrast to the fsRNNs, interference was largely concentrated on nearest neighbor intervals with changes to non-targeted intervals being opposite in sign to the target. We note that this is inconsistent with our birdsong data (Fig. 1d).

**Time-keeping in feedforward networks**. Feedforward networks have been proposed as alternatives to RNNs for generating temporal patterns[1,34,35,40]. By organizing neurons into layers connected in a feedforward manner, these networks naturally map layer-specific neural activity to elapsed time[1]. In this section, we focus on feedforward networks with a single neuron per layer (Fig. 4a). This simplification allows us to study the constraints that flexible time-keeping imposes on feedforward network dynamics analytically. Using insights gained from this treatment, we then discuss more realistic feedforward networks in the next section.

Applying the orthogonality constraint (Eq. 1) to feedforward networks revealed that flexible time-keeping requires that synaptic weights be mapped to interval durations in a unique one-to-one manner (Methods and Supplementary Note 2). Consider an interval $I^{(\alpha)}$, the duration of which depends on activity in layers $\alpha-1$ and $\alpha$. In Methods and Supplementary Note 2, we prove that flexible time-keeping requires changes to the synaptic weight between layers $\alpha-1$ and $\alpha$, denoted here by $W_\alpha$, to affect only the duration of $I_\alpha$:

$$\frac{\partial I^{(\alpha)}}{\partial W_\beta} = 0, \text{ if } \alpha \neq \beta. \tag{5}$$

As we show in Methods and Supplementary Note 2, the one-to-one mapping constraint (Eq. 5) is satisfied if synaptic weight changes that alter the timing of the post-synaptic neuron's activity leave its shape and magnitude unaffected. This ensures that downstream neurons encoding non-target intervals shift their activity in time by the same amount, keeping non-target interval durations unchanged, thus satisfying Eq. (5). Note that Eq. 5 is naturally satisfied by synaptic connections between downstream layers, as these cannot affect activity in upstream layers.

To further illustrate the constraints that Eq. (5) imposes on dynamics, we use a simple and analytically tractable example of a feedforward network, a chain of integrate-and-fire neurons (Fig. 4b), showing how it fails or succeeds in flexible time-keeping. In this example, initial synaptic weights are chosen such that when the first neuron in the chain produces a single spike, downstream neurons in the chain propagate activity by producing a single spike each. This network satisfies the one-to-one mapping constraint (Eq. 5) (Methods, Eq. 10). If $W_\alpha$ is increased, the synaptic weight matrix changes along the gradient (Methods, Eq. 10), advancing the firing of the $\alpha^{th}$ neuron since it reaches spiking threshold sooner. However, as long as the neuron produces only one spike, the time it takes for the next neuron to spike will not change, keeping all subsequent intervals unchanged. Similarly, decreasing $W_\alpha$ causes the duration of interval $I^\alpha$ to increase, without affecting the others. Note that this is exactly the mechanism described in the previous paragraph: synaptic weight changes affect only the timing of the post-synaptic response, not its magnitude or shape. Therefore, constraint (Eq. 5) is satisfied and the interference matrix is diagonal.

However, there are limits to the timing flexibility exhibited by a chain of integrate-and-fire neurons. If synaptic weights decrease below a point where the excitation in the post-synaptic neuron no

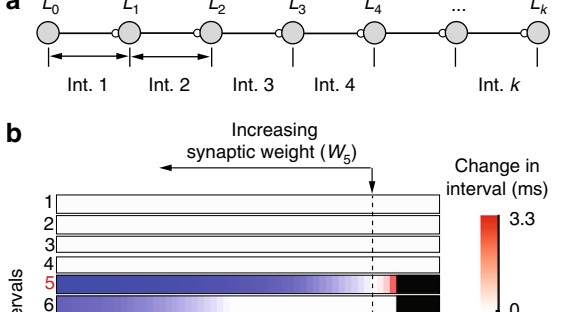

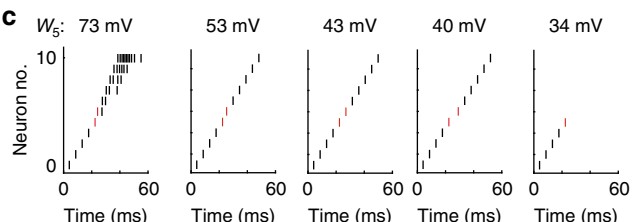

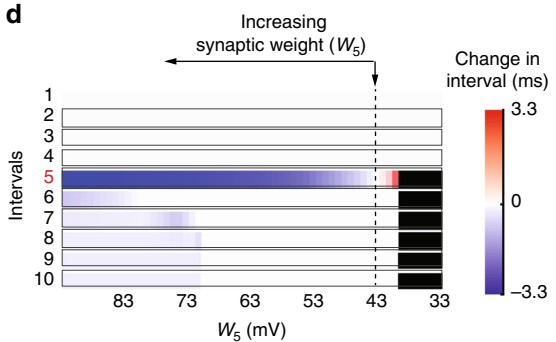

**Fig. 4** Flexible timing with chains of single neurons. **a** Schematic showing the network architecture. **b** A chain of 11 integrate and fire neurons was simulated with the first spike time of each neuron marking an interval boundary. $W_5$ (the weight between node 4 and 5) was varied, which corresponds to changes to the gradient vector for the fifth interval (Methods). Changes with respect to baseline interval durations, which were 4.6 ms, are shown. Black denotes chain propagation breakdown. Vertical arrow shows the baseline network configuration with each neuron spiking once. **c** Example runs of a chain of integrate and fire neurons. Only one neuron per layer is shown. **d** Same as **b**, except neurons have a 1 ms refractory period

longer drives it above spiking threshold, chain propagation stops (Fig. 4b). On the other end, if synaptic weights increase to a point where the post-synaptic neuron produces multiple spikes, then the downstream intervals may get shorter as the boost in excitation from the extra spike(s) will cause the downstream neurons in the chain to reach threshold sooner (Fig. 4b, c). In other words, if synaptic weight changes, beyond shifting the post-synaptic response in time, also alter its magnitude, this may interfere with downstream intervals. Therefore, timing flexibility holds only for a finite range of synaptic weights (see Supplementary Note 3 for the exact expressions). Importantly, this range can be increased by other mechanisms, such as refractoriness, that make the shape of the post-synaptic response,

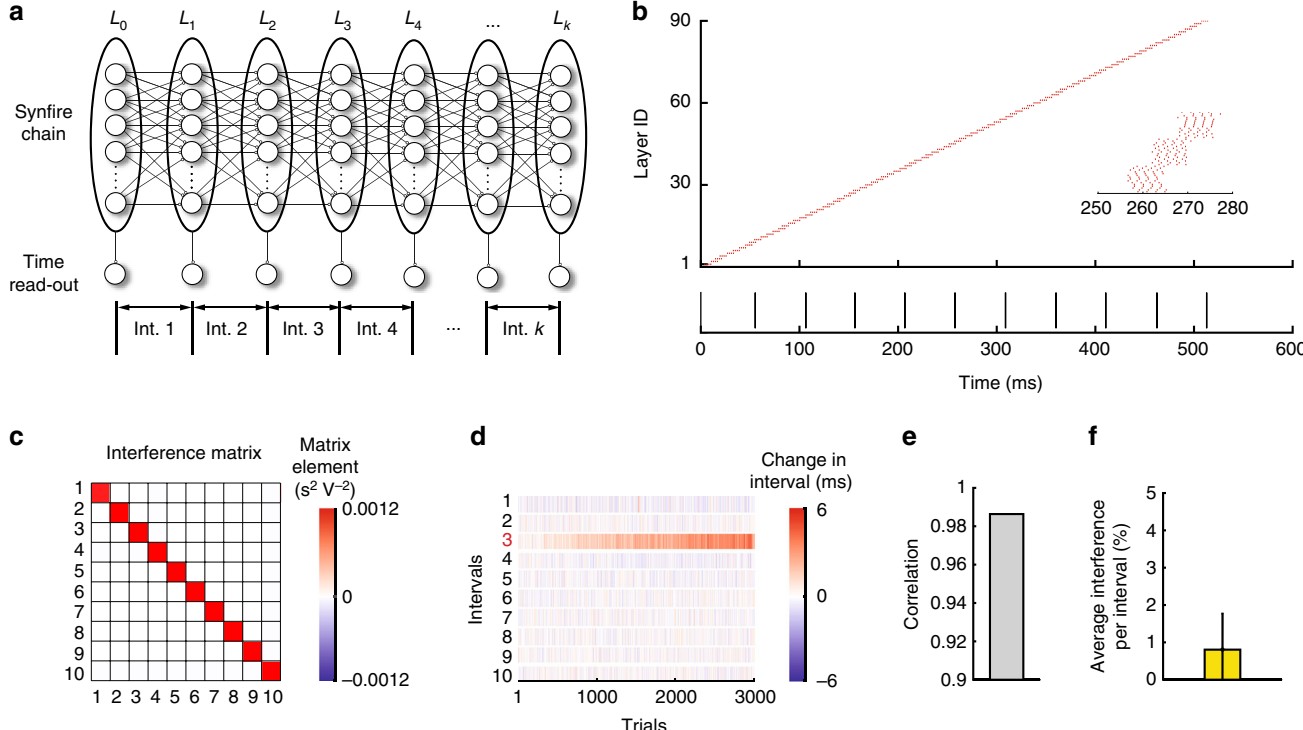

**Fig. 5** Flexible timing with biologically plausible synfire chains. **a** Schematic showing the network architecture and interval (Int.) time readout. **b** Example run of the synfire chain. Only one neuron per layer is shown. On the bottom is the activity of the read-out neurons that define the interval boundaries. **c** The interference matrix of the synfire chain. **d** Examples of a reinforcement learning 'experiment' that targeted interval 3 for lengthening. The effect on all intervals after 3000 trials is shown. **e** Correlation coefficient between the interval changes calculated from the interference matrix and from the reinforcement learning simulations (as in Fig. 2i). **f** Average interference per interval in reinforcement learning simulations, averaged across 20 learning simulations. Error bar shows standard deviation across simulations

number of spikes in our case, less sensitive to changes in input (Fig. 4d).

In realistic feedforward networks with multiple spiking neurons per layer, layer-to-layer propagation of excitation can occur in asynchronous[41,42] or synchronous modes[3,4]. It has been argued that the synchronous mode[4], is superior to the asynchronous mode[41] in terms of propagating activity across many layers, making it a viable solution for temporal pattern generation. Moreover, the synchronous mode, as we show next, can be made to conform to the constraints on flexible time-keeper networks.

**Synfire chains as flexible time-keepers**. We simulated a chain of integrate-and-burst spiking neurons, modeled on the putative time-keeper circuit of the songbird (HVC)[43–46]. The network had 90 layers with 15 neurons per layer, for a total of 1350 neurons (Fig. 5a, Methods), with the connectivity between layers being all-to-all. The network was configured to propagate activity in a synchronous mode, hence a 'synfire chain' (Fig. 5b). Output neurons received input from all neurons in every 9th layer. The first spike of each output neuron marked the time of an interval boundary (Fig. 5a, b), yielding 10 intervals of roughly 50 ms average duration. As expected from the analogy with the single-spike neuron chain, the interference matrix for the synfire chain was diagonal at this baseline configuration, i.e., non-diagonal elements were 0 (see Methods, Fig. 5c).

Modifying the duration of specific intervals using the same reinforcement learning algorithm as for the recurrent networks (Methods) resulted in learning rates comparable to what is seen in songbirds (Fig. 5d). During these simulations, synaptic weights

were bounded from above and below to ensure stable propagation through the chain. If synapses become too strong, neurons can produce more than single bursts leading to overexcitation, while if synapses become too weak activity can die out (Methods). Furthermore, such bounds prevent possible interference effects discussed above for the single neuron chain (Fig. 4b, c). Changes in the duration of targeted and non-targeted intervals were, on average, well predicted by the interference matrices (Fig. 5e). Importantly, the interference was an order of magnitude smaller than in the random recurrent networks, although not zero (~1%, Fig. 5f), and consistent with what we observed in songbirds (~0.7% when normalized to non-target segment duration in baseline). We verified that these results are robust to varying the the synfire chain model in at least two ways: (1) layer-to-layer connectivity was changed to all-to-all with weights being randomly chosen (Supplementary Fig. 4A, B and C), and (2) neurons made synapses to next layer neurons with some probability (Supplementary Fig. 4D, E and F).

To assess independence, we again targeted two non-adjacent intervals for simultaneous modification. The average reduction in learning rates was small (~3−5%) (Supplementary Fig. 4G and H), consistent with our observations in songbirds (Fig. 1g). These results suggest that synfire chains allow for flexible adjustments to motor timing.

**Variability in temporal structure**. To further delineate the network that implements song timing in zebra finches[43,47], we analyzed temporal variability of song segments targeted in our CAF paradigm. We found that when their intervals were shortened, variability in their durations decreased, whereas when

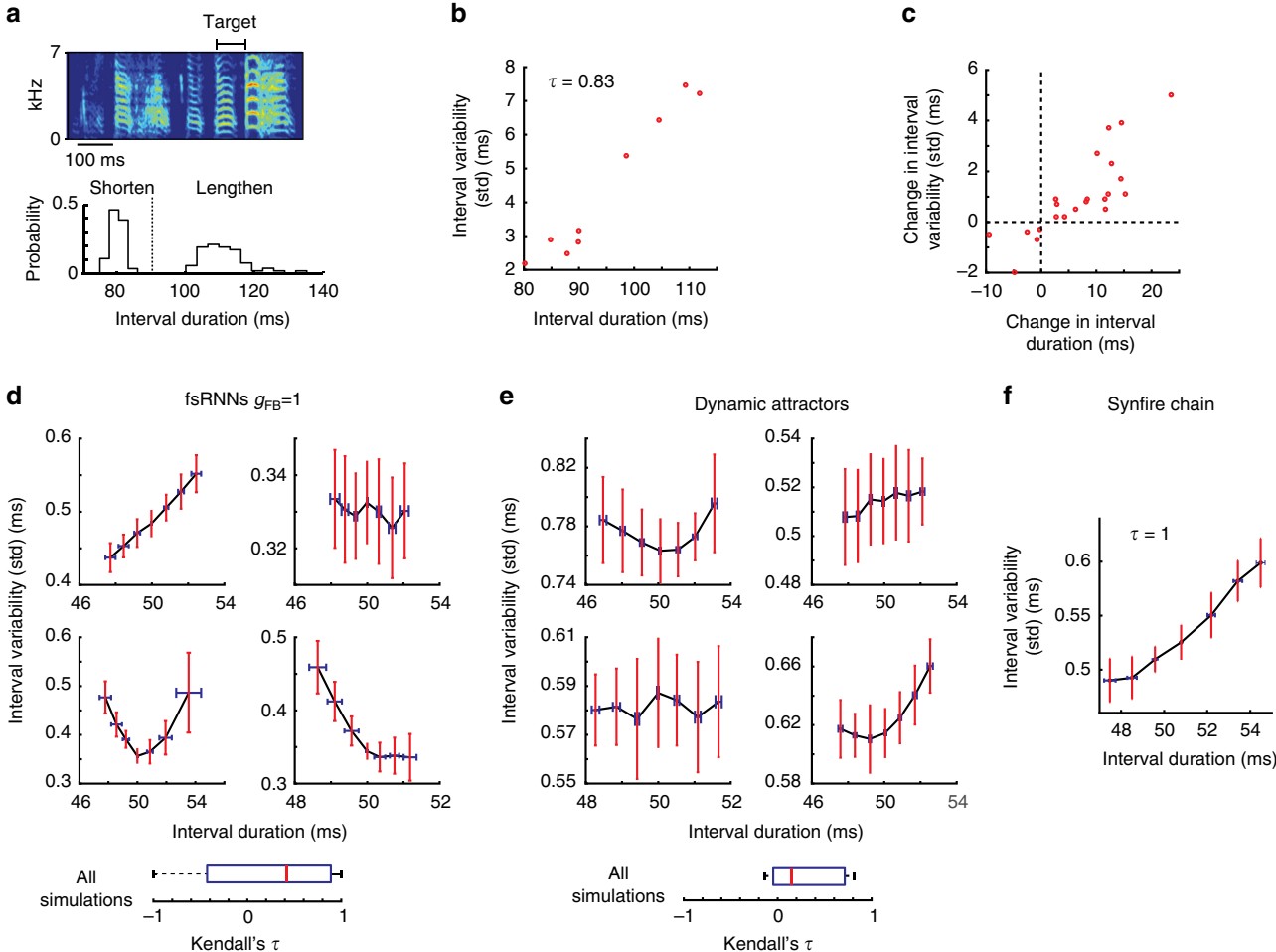

**Fig. 6** Changes in temporal variability with learning. **a** (Top) Spectrogram of a zebra finch song showing the target segment. (Bottom) Histograms of mean durations of a target segment after it had been shortened and lengthened using our CAF paradigm. Mean at baseline is denoted by dotted line. **b** Scatterplot of variability (standard deviation) as a function of mean segment duration for the same example as in **a** across 9 days of CAF (Kendall's $\tau =$ 0.83, $p = 8.5 \times 10^{-4}$, $n = 9$ days). **c** Scatterplot of change in variability (standard deviation) as a function of change in mean segment duration (both relative to baseline before CAF). **d** Standard deviation of target duration as a function of the target's mean duration for fsRNNs with $g_{FB} = 1$. For this value of feedback, the temporal variability is consistent with variability observed in zebra finch song (see the discussion in Methods). Target intervals were lengthened and shortened from baseline for 3000 trials each. Data from 200 'catch' (i.e., non-reinforced) trials interspersed between every 1000 trials, averaged across 20 simulations. Example from four fsRNNs are shown. Error bars are standard deviations across simulations. (Bottom) Box plot of all Kendall's $\tau$ coefficients between interval variability and interval duration, across 20 networks, 2 target intervals each (3rd and 8th). **e** Same as **d** for dynamic attractor networks. Box plot of all Kendall's $\tau$ coefficients across 14 networks, 1 target interval each (3rd). **f** Same as **d**, but for synfire chains. Data from 200 'catch' trials interspersed between every 1000 trials, averaged across 20 simulations. Target interval: 3rd

segments were lengthened their variability increased (Fig. 6a). This monotonic relationship held across multiple days of learning for the same target (Fig. 6b) and across our population of birds (Fig. 6c).

We analyzed the extent to which the network models we have considered recapitulate the monotonic relationship between changes in the mean duration and the variability of targeted segments. Neither of the recurrent pattern generator networks showed a consistent relationship between target interval duration and variability. Depending on initial network configuration, variability could increase, decrease, or have no correlation with changes in interval duration. It could even have a convex shape (Fig. 6d, e). In contrast, synfire chain networks showed an increase in variability as a function of changes in target interval duration (Fig. 6f and Supplementary Fig. 4C and F), conforming to the experimental observations in songbirds.

## Discussion

Modifying the temporal structure of stereotyped motor sequences is an important means of adapting motor output to new task demands[12–14,19]. Here we combined behavioral experiments in songbirds with theory and network simulations to explore the flexibility with which pattern generator circuits, real and simulated, can modify the temporal structure of their output. We found that the timing of zebra finch song can be altered with very high degree of specificity (Fig. 1), and derived formal prescriptions for neural networks implementing such flexible motor timing.

We show that gradient vectors in synaptic weight space associated with different intervals in a sequence must be orthogonal for the intervals to be independently modifiable. Certain feedforward networks generally satisfy this criterion making them ideally suited as flexible time-keepers. The general topology of RNNs, however, does not guarantee such flexibility. Whereas synapses in feedforward networks naturally map onto specific

time points in a sequence, synapses in RNNs that contribute to changing one interval are often not unique from those that drive changes in other intervals, leading to interference in the time domain.

We found that the degree of timing flexibility further depends on the specifics of the networks and the properties of its neurons. In RNNs, strong feedback from output neurons to the RNN, which can increase the network's robustness to perturbations (Fig. 2d), makes the network less flexible in the temporal domain (Fig. 2f). The dynamic attractor network of Laje and Buonomano[7], which does not require any feedback, shows overall less interference (Fig. 3). While feedforward networks are structurally better suited, we found that flexible time-keeping is compromised in these networks if the synaptic changes between neurons alter post-synaptic spiking responses beyond just shifting them in time. In this regime, modifying the strength of a synapse will not only affect the interval associated with that synapse, but the effect will propagate to downstream neurons causing changes also to subsequent intervals (Supplementary Notes 2 and 3).

The range over which feedforward networks operate as flexible time-keepers can be extended by making its neurons' spiking responses less sensitive to the magnitude of pre-synaptic input. Interestingly, projection neurons in HVC, the putative time-keeper circuit in songbirds[16,43,48], generate calcium bursts that accomplish just that. Modeling a synfire chain network with integrate-and-burst neurons, akin to those described in HVC[43], recapitulated the flexible timing observed in songbirds (Fig. 5).

Additional support for the idea that song timing is governed by a synfire chain-like network comes from the relationship between variability in interval duration and changes to its mean (Fig. 6a −c). Whereas RNNs show no consistent relationship (Fig. 6d, e), feedforward networks predict the monotonically increasing relationship between variability and mean duration we observe (Fig. 6f). This relationship is a consequence of how interval durations are altered in feedforward networks: synaptic strengthening between neurons in subsequent layers makes the signal propagate faster, shortening the associated interval. But synaptic strengthening also makes the synapse less prone to noise and hence less variable[49], while weakening the synapse has the opposite effects. Whether this characteristic monotonic relationship between interval duration and variability, suggestive of an underlying feedforward network architecture, is seen also in other flexible behaviors, such as speech, remains to be investigated[22].

While both random recurrent and feedforward neural networks have been proposed to underlie the temporal structure of birdsong[47,48], our findings together with other recent experimental results[16,43], suggest that the network that controls song timing functions as a synfire chain. The extent to which the synfire chain-like network is implemented locally in HVC, or in a more distributed network[50], remains to be further explored. Importantly, our results add intuition as to why a synfire chain-like network architecture is the preferred solution: it allows very specific changes to the song's temporal structure.

We also found that modifications to interval durations in RNN networks result in more temporal interference than predicted by gradient ascent (Figs. 2k, 3i). This is because individual learning experiments do not find the optimal synaptic updates, i.e., those that align with the gradient vector for the targeted interval. This solution only emerges when averaging across many experiments. For an intuition as to why trial-and-error learning leads to more interference, we can consider two intervals whose gradients are orthogonal, one of which is targeted for modification. Because the synaptic updates at the end of an experiment will not exactly point along the target interval gradient, the updates will not be orthogonal to the gradient of the non-targeted interval and will hence cause a change in its duration too, i.e., interference.

While we have limited our study to structurally feedforward networks, our results generalize also to functionally feedforward networks[34]. In these networks, specific modes of network activity (Schur modes), rather than individual neurons, are organized into layers and the connectivity matrix is feedforward in the basis of these modes. Our results suggest that timing flexibility in such networks would be compromised if synaptic changes alter the Schur modes beyond just shifting them in time. What is needed for flexible time-keeping is a 'generalization' of the synfire chain to a functionally feedforward architecture, although how such generalizations can be made is not clear. Asymmetric Hopfield networks[6,8] might provide insight into how to construct such networks. Even though these networks are built with binary neurons, they describe dynamics that evolve from one attractor to another and their connectivity has a feedforward structure in the basis of these attractors. We speculate that the attractor states of the asymmetric Hopfield networks might be analogous to synchronous spiking of neurons in a synfire chain.

One could also consider hybrid architectures, where separate RNNs encode the individual segments of a longer sequence (e.g., song) and are then coupled in a feedforward manner. If learning-induced modifications to the feedforward connections only shift the time at which the next recurrent network starts its activity, this architecture would also function as a temporally flexible pattern generator.

Modifications to timing could also be induced by changing the spatiotemporal profile of the inputs to a pattern generator circuit, as opposed to synaptic plasticity within the pattern generator as considered here. This amounts to pushing the problem we have discussed upstream of the pattern generator, since it is the network providing the input that must now ensure that timing is modified flexibly. Interestingly, synfire chains provide a simple way for upstream networks to modify timing in a flexible manner. Suppose all neurons in a layer get a common tonic input from an upstream area, and these inputs are independent from layer to layer. By increasing or decreasing the tonic input a layer gets, without changing its temporal profile, the upstream area can make the neurons in the layer fire earlier or later, without interfering with other interval durations. Modifying timing by means of changing the spatial, but not temporal, profiles of inputs is made possible by the one-to-one mapping of time to space inherent to synfire chains. The described mechanism, in addition to flexible learning, may also allow flexible real-time control of motor timing. For example, in Bengalese finches, acute perturbations of auditory feedback led to slowing down of the target syllable duration[51,52] accompanied by a decrease in HVC activity[52]. It is conceivable that such decrease in activity is due to a change in auditory feedback-related input to HVC. It would be interesting to quantify how specific the tempo changes are to the target syllable in these experiments.

In this study, we only discussed flexibility in motor timing. However, depending on the structure of the network controlling the behavior, temporal changes could also interfere with spatial aspects of motor output[1]. In the songbird system, for example, the output neurons of the time-keeper network, HVC, synapse onto motor cortex analog neurons that, indirectly, drive the vocal muscles[25]. If modifications to the song's temporal structure change the magnitude of the spiking response in the output neurons, this could change not only the timing of muscle activations but also their magnitude. Thus separating the temporal and spatial aspects of movements[19,23] requires separating the timing of an output neuron's firing from its magnitude. In general, neither RNNs nor feedforward networks ensure such a separation. However, synfire chains do, suggesting another reason why they might have the ideal topology and dynamics for implementing the timing of flexible behaviors.

Maximal flexibility in modifying motor skills would analogously require the capacity to change specific movement features without interfering with others. The same requirements for flexibility in motor timing, i.e., independence and flexibility, also apply to spatial aspects of movements. For example, in a reinforcement learning model of a reaching task[53], interference between different movements were likely[54], significantly slowing down the learning process. The methods and formalisms we present for understanding the network constraints associated with flexible motor timing can be extended to study networks underlying flexibility in the spatial domain.

Beyond informing the neural circuits underlying flexible time keeping, our study is also an example of how behavioral experiments in combination with network modeling and theory can inform the structure and dynamics of neural circuits underlying complex behaviors.

## Methods

**Birdsong experiments**. Animals: The care and experimental manipulation of all animals were reviewed and approved by the Harvard Institutional Animal Care and Use Committee. Experimental subjects were adult male zebra finches (>90 days post-hatch, $n = 24$).

Behavioral experiments: Adaptive changes to the duration of targeted song segments were induced as previously described[19]. Briefly, we computed the duration of target segments in real-time using a static threshold crossing of the smoothed amplitude envelope (5 ms boxcar filter with 1 ms advancement). If the duration did not meet the threshold, a burst of white noise (80−90 dB) was played through a loudspeaker for 50−100 ms with short latency (~1−3 ms). Syllable onsets are associated with rapid increases in sound amplitude, which makes the estimates of their timing more robust to noise. Thus, we mostly targeted 'syllable +gap' segments and estimated the target segments from the onset of the target syllable to the onset of the following syllable. In a typical single-target experiment, birds underwent 3−5 days of CAF to lengthen the target followed by 3−5 days of CAF to shorten the target (for a total of 6−10 days for each experimental block) and no CAF for at least 4 days before subsequent experiments.

When comparing learning rates across experiments in which one ("alone", Fig. 1f-g) versus two targets ("with other") were targeted for modification, we chose targets that were separated by at least 100 ms. We first ran the CAF protocol on a randomly selected group of animals ($n = 3$ birds) for target 1, then for target 2, before targeting both intervals in the same experiment. In another group of birds ($n = 3$), we counterbalanced the order, running the two-target experiments first, followed by single-target experiments. The two targets in the same song were modified in opposite directions.

Birdsong analysis: All analyses of learning rates and variability of segment durations were done offline on "catch trials" during which the white noise feedback was turned off for up to 100 song renditions in the morning (AM) and again approximately 8 h later in the evening (PM). The method for obtaining estimates of song segment durations has previously been described[19]. To calculate learning rates, we computed the change in the target duration from the start to the end of a CAF run (up or down) and divided it by the number of intervening days. We compared both morning and evening catch trials, averaging across them to obtain a more robust estimate of learning rate. We compared variability in interval duration for the same time-period in the day in order to rule out potential circadian effects.

To compare the changes in segment durations during CAF to normal baseline drift, we used the same number of days of baseline (before CAF was started) as the number of days of CAF, which for each bird varied between 6–10 days. Note that for all changes in duration (CAF or baseline), they are divided by number of days (ms/day) and not absolute total change. We subtracted the absolute value of this baseline drift from the CAF-induced changes to obtain a better estimate of learning. We did the same for non-target intervals, discarding the signs for each non-target interval change when averaging within a bird to rule out that different signs negated each other, thus masking any potential non-target effects.

For Fig. 1g, we analyzed the data as follows to prevent pseudoreplication. For each bird, we averaged the change/day for the 2 segments in the alone condition and did the same for the w/other condition. This produced a vector of 6 numbers (one for each bird) for each of the two conditions. Because the same birds were used in both conditions, we then did a paired samples *t*-test between the two conditions

Statistical analysis: No explicit power analysis was conducted to predetermine sample sizes. However, our sample sizes were generally similar to those used in previous birdsong studies. We used two-sided statistical tests as noted in the Figures. All statistics presented in the main text refer to mean ± SD, while error bars in the figures all represent SEM.

**Theoretical and computational methods**. Flexibility of pattern generator circuits: A detailed account of the constraints imposed on pattern generator circuits for

flexible time-keeping is given in Supplementary Note 1. Here, we present simple derivations of Eqs. (1−4).

Gradient ascent on reward, which we assume to be contingent on the duration of interval $\alpha$, requires changes in the synaptic weights along the reward gradient, which is related to the interval duration gradient by the chain rule:

$$\frac{dW_{ij}^{\alpha}}{dl} = \eta \frac{\partial R^{\alpha}}{\partial W_{ij}} = \eta \frac{dR^{\alpha}}{dI^{\alpha}} \frac{\partial I^{\alpha}}{\partial W_{ij}}, \tag{6}$$

where $\eta$ is a learning rate parameter, $\mathbf{W}^{\alpha}(l)$ is the trajectory in synaptic weight space traced during gradient ascent, and $l$ parametrizes the trajectory. Changes in the duration of an interval along the curve $\mathbf{W}^{\alpha}(l)$ is given by:

$$\frac{dI^{\beta}}{dl} = \sum_{\{ij\}} \frac{\partial I^{\beta}}{\partial W_{ij}} \frac{dW_{ij}^{\alpha}}{dl}, \tag{7}$$

where the summation is over existing (non-zero) synaptic weights. Eq. (6) can be used to substitute the last term in Eq. (7) to give:

$$\frac{dI^{\beta}}{dl} = \eta \frac{dR^{\alpha}}{dI^{\alpha}} \sum_{\{ij\}} \frac{\partial I^{\alpha}}{\partial W_{ij}} \frac{\partial I^{\beta}}{\partial W_{ij}}. \tag{8}$$

Eq. (8) shows that unless Eq. (1) is satisfied, non-targeted intervals change while gradient-ascent is performed.

For small changes in synaptic weights, one can approximate the changes in intervals using a linear approximation: $\delta I^{\beta} \approx \frac{dI^{\beta}}{dl} l = \eta \frac{dR^{\alpha}}{dI^{\alpha}} \sum_{\{ij\}} \frac{\partial I^{\alpha}}{\partial W_{ij}} \frac{\partial I^{\beta}}{\partial W_{ij}} l = \eta \frac{dR^{\alpha}}{dI^{\alpha}} M_{\alpha\beta} l$.

Eq. (3) directly follows from Eq. (8) by substitution.

A more general scenario is when there are multiple reinforcers acting on multiple intervals simultaneously. Total reinforcement is now given by $R = \sum R^{\alpha}$. In this case, the interference patterns can be more elaborate, however it is still governed by the interference matrix:

$$\delta I^{\beta} \approx \eta \sum_{\alpha} \frac{dR^{\alpha}}{dI^{\alpha}} M_{\alpha\beta} l. \tag{9}$$

For example, Eq. (4) can be derived using similar arguments to derivation of Eq. (3), except now the reinforcement is $R^{\alpha} + R^{\beta}$.

FsRNNs. Network setup and training: We simulated RNNs with $N = 500$ neurons. The neural dynamics was described by,

$$\tau_r \frac{dx_i}{dt} = -x_i + \sum_{j=1}^{N} W_{ij} r_j(t) + \sum_{j=1}^{2} W_{ij}^{\text{In}} y_j(t) + g_{\text{FB}} W_i^{\text{FB}} z(t) + \sqrt{\tau_{\eta}} \sigma \eta_i(t),$$

$$r_i(t) = \tanh(x_i(t)),$$

$$z(t) = \sum_{j=1}^{N} W_{ij}^{\text{Out}} r_j(t),$$

where $i = 1, \ldots, N$, $r_i(t)$ are the firing rates of the neurons, $y_j(t)$ are the inputs, and $z(t)$ is the output. The parameters of the network followed[9], with the neural time constant being $\tau_r = 10$ ms. The recurrent synaptic connectivity matrix, $\mathbf{W}$, was sparse: $W_{ij}$ was 0 with probability $1−p$, with $p = 0.1$. Non-zero elements of $\mathbf{W}$ were drawn independently from a Gaussian distribution with 0 mean and a variance of $1.5/(pN)$, ensuring that the network is in the chaotic regime without the feedback from the output[10]. Elements of $\mathbf{W}^{\text{In}}$ and $\mathbf{W}^{\text{FB}}$ were independently drawn from a uniform distribution between −1 and 1. The strength of feedback $g_{\text{FB}}$ was varied, as discussed in the main text. Unless stated otherwise, the networks were simulated with independently injected noise to neurons, including during training: $\eta_i(t)$ is a zero-mean Gaussian white noise with covariance $\langle \eta_i(t)\eta_i(t') \rangle = \delta(t - t')$, $\tau_{\eta} = 10$ ms and $\sigma = 0.01$.

The noise magnitude was chosen to ensure that after training, the coefficients of variation of temporal variability in the interval durations were about 0.5 ms, consistent with pattern generator variability in zebra finch song[28]. Variability in song tempo is comprised of temporal variability in the pattern generator circuits and 'peripheral jitter'. Jitter can be caused by temporal variability in downstream circuits and muscles[28] and is estimated to contribute 2 ms of variability on average, independent of interval duration[28]. Combined with the 0.5 ms variability observed in our models for a 50-ms interval, total variability would add up to $2.06 \, \text{ms} = \sqrt{(0.5 \, \text{ms})^2 + (2 \, \text{ms})^2}$. This corresponds to a coefficient of variation (CV) of ~4% for song timing, consistent the ~1–10% CV range observed in young adult zebra finches (see refs. [28,55,56] and Fig. 6b).

There were two types of inputs to the network: (1) $y_1(t)$ was a unit pulse of duration 50 ms and amplitude 5 (A.U.). This input was present for all simulations. The end of the pulse marked the beginning of the first interval. (2) $y_2(t)$ was a unit pulse of duration 10 ms and amplitude $P$ that was varied. It arrived 120 ms after the start of the first interval. This input was used only when assessing the robustness of the network to perturbations (Fig. 2d).

The desired output was a 530-ms waveform (Supplementary Fig. 1A) constructed by:

(1) summing 10 Gaussian waveforms centered 50 ms apart with widths (standard deviation) of 10 ms,

(2) normalizing the waveform such that its maximum is 1 and minimum 0.1, and

(3) shifting it in time such that the first threshold-crossing (which is chosen to be 0.68) occurs 50 ms after the first interval.

Hence, the desired output marked 10 equally spaced 50 ms intervals. $W^{\text{Out}}$ was trained using the FORCE algorithm[9] for 30 training trials. After training, 10 test runs were performed and the error between the network output and the desired output ('test error') calculated for each test run. The test error was defined by

$$\frac{\sqrt{\int_0^{530\,\text{ms}} (z_{\text{des}}(t) - z(t))^2 \, dt}}{\sqrt{\int_0^{530\,\text{ms}} z_{\text{des}}(t)^2 \, dt}}, \text{ where } z_{\text{des}}(t) \text{ was the desired output.}$$

The equations governing network dynamics were integrated with a first-order Euler method. The integration step size was $dt = 0.1$ ms, unless otherwise stated.

For Fig. 2c, the initial state distance between two intervals was calculated by first making two $N = 500$ dimensional vectors of the instantaneous firing rates of the network neurons at the beginning of the two intervals and then calculating the $L_2$-norm difference of these vectors.

Interference matrix. To calculate the interference matrices of trained fsRNNs, we calculated the gradients of interval durations numerically: one-by-one, each non-zero synaptic weight was increased by $dW = 0.05$ and the change in interval durations measured. In these simulations, $\sigma = 0$ and $dt = 0.01$ ms. To check the accuracy of our results, we ran the same calculation with $dt = 0.1$ ms and observed that changes in our reported values were small, e.g., less than 1% in Fig. 2f.

Reinforcement learning. We implemented a modified version of the synaptic plasticity rule of Fiete and Seung[31] to adaptively change the non-zero elements of $W$. Specifically, changes to synaptic weight $W_{ij}$ were given by

$$\Delta W_{ij} = \gamma \int_0^{t^{(K)}} R(t) e_{ij}(t) \, dt,$$

where $t^{(K)}$ denotes the end of the last interval, $\gamma = 0.004$, $e_{ij}(t)$ the eligibility trace, and $R(t)$ the reward. $\gamma$ was chosen to match learning rates observed in experiments with songbirds. The eligibility trace is:

$$e_{ij}(t) = \int_0^t \frac{dt'}{\tau_e} e^{-(t-t')/\tau_e} \eta_i(t') r_j(t'),$$

where $\tau_e = 35$ ms[19]. The reward signal is given by $R(t) = R^{(\text{tar}_1)} \delta(t - t^{(\text{tar}_1)})$ in a single-target interval 'experiment' and by $R(t) = R^{(\text{tar}_1)} \delta(t - t^{(\text{tar}_1)}) + R^{(\text{tar}_2)} \delta(t - t^{(\text{tar}_2)})$ in a two-target intervals 'experiment', where $t^{(\text{tar}_1)}$ and $t^{(\text{tar}_2)}$ denote the times at which the 1st and 2nd target intervals end, and $R^{(\text{tar}_1)}$ and $R^{(\text{tar}_2)}$ are 0–1 rewards contingent on 1st and 2nd target intervals respectively. If the interval is not targeted, the reward is always 0. If the interval is targeted for modification, the rewards are calculated by comparing the interval duration in the current trial, $I^{(\text{tar})}$, to the running average of the target interval duration, $\bar{I}^{(\text{tar})}$, which is updated in each trial as follows: $\bar{I}^{(\text{tar})} \leftarrow 0.995 \bar{I}^{(\text{tar})} + 0.005 I^{(\text{tar})}$. If the interval is targeted for lengthening, the reward is 1 if $I^{(\text{tar})} > \bar{I}^{(\text{tar})}$, and 0 otherwise. If the interval is targeted for shortening, the reward is 1 if $I^{(\text{tar})} < \bar{I}^{(\text{tar})}$, and 0 otherwise.

In Supplementary Fig. 2F-G, we simulated a single-interval experiment with an internally generated reinforcer, which preserves the baseline (or template) values of interval timings, $\bar{I}_{\text{temp}}^{(i)}$, which is calculated by averaging of 400 trials without reinforcement. The total reward is given by

$$R(t) = R^{(\text{tar}_1)} \delta(t - t^{(\text{tar}_1)}) + \sum_{i=1}^{10} R_{\text{temp}}^{(i)} \delta(t - t^{(i)}). \text{ Here } R_{\text{temp}}^{(i)} \text{ is the template}$$

reinforcement signal. $R_{\text{temp}}^{(i)} = 0.2$ if $\left| \bar{I}^{(i)} - \bar{I}_{\text{temp}}^{(i)} \right| > 0.1$ ms and $\left( \bar{I}_{\text{temp}}^{(i)} - \bar{I}^{(i)} \right) \left( I^{(i)} - \bar{I}^{(i)} \right) > 0$, $R_{\text{temp}}^{(i)} = 0$ otherwise.

Interval duration changes during a reinforcement learning experiment are calculated by a running (200 point window) across trials and subtracting from them their baseline values. The baseline values are averages of the interval durations across 400 trials where no reinforcement was delivered.

Dynamic attractor. Network setup and training: The architecture of this network, the dynamics of its neurons and the numerical integration of the dynamical equations, its connectivity and noise parameters, as well as its input and output waveforms, are the same as the fsRNNs, except the feedback from the output was set to zero, i.e., $g_{\text{FB}} = 0$.

The networks are trained using the 'innate training' procedure[7], which consisted of two stages:

(1) 70% of non-zero elements of $W$ are trained using the FORCE algorithm (30 training trials) to stabilize an innately produced trajectory of duration 530 ms. See ref. [7] for details.

(2) $W^{\text{Out}}$ is trained using the FORCE algorithm to produce the desired output waveform, exactly as in fsRNNs.

Interference matrix. The procedures for calculating the interference matrices are the same as for the fsRNNs.

Reinforcement learning. The procedures for reinforcement learning are the same as the fsRNNs, except that $\gamma = 0.004/3$.

Chains of single neurons. One-to-one mapping of synaptic weights to intervals: In Supplementary Note 2, we prove that in a feedforward network with a single neuron per layer flexible time-keeping requires that the $\alpha^{\text{th}}$ interval depend only on the synaptic weight between layers $\alpha - 1$ and $\alpha$, $W_\alpha$. To prove this result, we make the following general assumptions: Activity in each layer codes for the start of an interval and end of the previous one. The network dynamics and the time read-out for each layer is such that: $t^{(\alpha)} > t^{(\beta)}$ when $\alpha > \beta$, where $t^{(\alpha)}$ is the time read-out from $\alpha^{\text{th}}$ layer. Finally, only the initial layer receives external input.

Chain of single integrate-and-fire neurons. We consider a chain of leaky integrate-and-fire neurons, which integrate their synaptic input currents with a leak time constant of $\tau_{\text{IF}}$. We assume that there is a non-zero volume of synaptic weight space in which the activity propagates in the chain with each neuron producing a single spike and the synaptic weights of the network configured to be in that volume (and not at the boundary of the volume where an infinitesimal change in synaptic weights may lead to multiple spikes per neuron or no spikes at all). The spike time of the neuron in the $\alpha^{\text{th}}$ layer is denoted by $t^{(\alpha)}$. The subthreshold dynamics of the neuron's membrane potential is given by:

$$\tau_{\text{IF}} \frac{dV_\alpha}{dt} = -(V_\alpha - V_{\text{rest}}) + W_\alpha E\left(t - t^{(\alpha-1)}\right),$$

where $\tau_{\text{IF}}$ is the leak time constant, $W_\alpha$ is the synaptic weight between layers $\alpha - 1$ and $\alpha$, and $E(t)$ is the causal post-synaptic potential that follows a pre-synaptic spike. When the membrane potential reaches a threshold, $V_{\text{th}}$, the neuron produces a spike and the membrane potential is reset to $V_R$. We assume that the neuron is at rest potential, $V_{\text{rest}}$, when the pre-synaptic spike arrives. The $\alpha^{\text{th}}$ neuron spikes when its membrane potential reaches the threshold:

$$V_{\text{th}} = V_{\text{rest}} + W_\alpha \int_{t^{(\alpha-1)}}^{t^{(\alpha)}} \frac{dt}{\tau_{\text{IF}}} e^{\frac{t^{(\alpha)} - t}{\tau_{\text{IF}}}} E\left(t - t^{(\alpha-1)}\right),$$

Taking the derivative of this equation with respect to synaptic weights and noting that $I^{(\alpha)} = t^{(\alpha)} - t^{(\alpha-1)}$, we get:

$$\frac{\partial I^{(\alpha)}}{\partial W_\beta} = \begin{cases} -\frac{\tau_{\text{IF}}}{W_\alpha \left( \frac{W_\alpha E\left(I^{(\alpha)}\right)}{V_{\text{th}} - V_{\text{rest}}} - 1 \right)} & \alpha = \beta \\ 0 & \alpha \neq \beta \end{cases}. \tag{10}$$

In scenarios where some neurons spike more than once, this result still holds for intervals, $I^{(\alpha)}$, for which the $(\alpha - 1)^{\text{th}}$ neuron spikes only once.

In simulations shown in Fig. 4, the following parameters were used: $\tau_{\text{IF}} = 10$ ms, $V_{\text{th}} = -50$ mV, $V_{\text{rest}} = V_R = -60$ mV, $W_\alpha = 43$ mV. $E(t) = \Theta(t) e^{-t/\tau_s}$ with $\Theta(t)$ being the step function and $\tau_s = 5$ ms. For Fig. 4c, a 1 ms refractory period followed a spike during which the membrane potential was set to $V_R$. The dynamical equations governing the network dynamics were integrated with a first-order Euler method, with an integration step size of $dt = 0.01$ ms

Synfire chain. Network setup and training: Our synfire chain model consisted of 1350 integrate-and-burst neurons organized into 90 layers, with 15 neurons in each layer (Fig. 4a). Neurons in a layer projected to all neurons in the next layer, forming a chain topology. The subthreshold membrane potential of the $i^{\text{th}}$ neuron, $V_i$, obeyed

$$\tau_{\text{IF}} \frac{dV_i}{dt} = -(V_i - V_{\text{rest}}) + I_{\text{syn},i}(t) + I_{\text{ext},i}(t) + \sqrt{\tau_\eta} \sigma \eta_i(t),$$

where $\tau_{\text{IF}} = 10$ mV, $V_{\text{rest}} = -60$ mV, $\eta_i(t)$ is a zero-mean Gaussian white noise with covariance $\langle \eta_i(t) \eta_i(t') \rangle = \delta(t - t')$, $\tau_\eta = 10$ ms, and $\sigma = 2$ mV. The synaptic inputs are given by $I_{\text{syn},i}(t) = \sum_j W_{ij} \sum_k E\left(t - t_j^k\right)$, where $t_j^k$ denotes the $k^{\text{th}}$ spike of $j^{\text{th}}$ neuron, $E(t) = \Theta(t) e^{-t/\tau_s}$ with $\Theta(t)$ being the step function and $\tau_s = 5$ ms, and $W_{ij}$ is 1.13 mV for synapses from a neuron to the neurons in the next layer and 0 otherwise. When the membrane potential of the integrate-and-burst neuron reaches threshold, $V_{\text{th}} = -50$ mV, the neuron emits 4 spikes with 2 ms between spikes and the membrane potential resets to $V_R = -55$ mV after a refractory period of 4 ms. Chain propagation starts by $I_{\text{ext},i}(t)$, a 5-ms pulse input with magnitude 30 mV applied only to neurons in the first node.

In Supplementary Fig. 4A and B, $W_{ij}$ were chosen uniformly in the range 1–1.27 mV. In Supplementary Fig. 4C and D, $W_{ij}$ was 1.1425 mV with probability 0.8 for synapses from a neuron to the neurons in the next layer and 0 otherwise. To map network dynamics to timing, an integrate-and-fire readout neuron is connected to all the neurons in every 9th layer, making a total of 10 readout neurons. Their dynamics are governed by the same equations as the chain neurons, except they do not receive any external input or noise. Synaptic weights between neurons in the chain were set to 1.13 mV. The first spikes of the output neurons mark interval boundaries. The first interval commences with the start of the external pulse.

The equations governing the network dynamics were integrated with a first-order Euler method, with an integration step size of $dt = 0.1$ ms, unless stated otherwise.

**Interference matrix.** To calculate the interference matrices of the synfire chain, we calculated the gradients of the interval durations numerically: one-by-one, each non-zero synaptic weight was increased by $dW = 0.113\,V$ and the change in interval durations measured. In these simulations, $\sigma = 0$ and $dt = 0.5$ μs. We observed that when a synaptic weight is increased by $dW$, durations of the interval associated with the perturbed synapse, decreased by 0.2645 ms, giving a gradient of 2.3407 ms/V. Changes in other intervals were numerically 0. A zero gradient is also expected from theory. Since our model synfire chain has all-to-all connectivity between layers and, at setup, all synaptic weights are the same, in the absence of noise, neurons that live in the same layer receive exactly the same input. A small increase in a synaptic weight leads to a larger synaptic input to all post-synaptic layer neurons, and causes them to shift their bursting to an earlier time by an equal duration, without changing the shape of the burst. Subsequently, downstream layers of the chain only shift their activity in time by equal durations, leading to no change in interval durations.

**Reinforcement learning.** The procedures for reinforcement learning were kept the same as for the fsRNNs, except that $\gamma = 1$ μV with the eligibility trace now taking the form:

$$e_{ij}(t) = \int_0^t \frac{dt'}{\tau_e}\, e^{-(t-t')/\tau_e}\, \eta_i(t')\, s_j(t'),$$

where $\tau_e = 35$ ms, and $s_j(t) = \sum_k E\left(t - t_j^k\right)$. The synaptic weights were not allowed to increase above $1.6$ mV nor fall below $0.92$ mV for stability of chain propagation: weights above the upper bound led to explosion, and weights below the lower bound led to activity propagation terminating before reaching the end of the chain.

**Statistics:** In figures, all reported error bars are standard deviations over different trained networks unless stated otherwise. In Figs. 2k, 3i, 5f, interference per interval, for a given network and a given reinforcement learning simulation of 1100 trials, was calculated as follows: (1) Intervals whose mean duration had changed were identified. For this purpose, for each interval, we performed a t-test (significance value 5%) to test whether the interval's durations between trials 900 and 1100 was coming from a distribution whose mean is the interval's mean baseline duration. (2) When the null-hypothesis was rejected (mean duration changed) for an interval, the mean duration at 1000th trial was estimated by averaging over the interval durations between trials 900 and 1100, and the change in mean duration was calculated by subtracting the baseline mean. When the null-hypothesis was borne out, the change in mean duration was zero. (3) The absolute change in non-target interval durations were normalized by the change in target duration and averaged over non-target intervals.

**Data and code availability.** All data analyzed and code used for this paper are available from the authors upon reasonable request.

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

## Acknowledgements

This work was supported by a Klingenstein Fellowship to B.P.Ö., Alzheimer's Association Research Fellowship to F.A. and partly by a Swartz Foundation post-doctoral fellowship to C.P. We thank Raoul Memmesheimer, Ben Dongsung Huh, and Brian DePasquale for discussions.

## Author contributions

B.P.Ö., C.P. and F.A. designed the study. C.P. performed the theoretical analysis and simulations with input from other authors. F.A. performed the birdsong experiments and analyzed the data with input from other authors. B.P.Ö. supervised all aspects of the study. B.P.Ö., C.P. and F.A. wrote the paper.

## Additional information

**Competing interests:** The authors declare no competing interests.

