## [Peer Review File · Nature Communications]

Reviewers' comments:

Reviewer #1 (Remarks to the Author):

Review of Pehlevan (#107871)

Summary:

In this manuscript, the authors integrate experimental and computational approaches to understand the neural mechanisms underlying the control of vocal timing. The authors provide elegant experiments on syllable timing that complement extensive tests of distinct computational models. In the end, the authors provide important support for synfire chain models of vocal motor control in songbirds.

Major points:

1. While the behavioral experiments are intriguing, they are less fleshed out than computational models. Currently, while the authors provide some data to allow comparisons between computational models and behavior, the behavioral analysis is much more superficial. For example, it is important to plot something akin to the interference (flexibility) matrices (e.g., Figure 2E) for the observed behavioral data. This will allow one to more directly compare the results from computational models with the observed data. This will also allow the reader to evaluate the degree to which the distance from the target interval could affect timing (see also comment below). Such a relationship has been demonstrated for the control of syllable structure (e.g., Hoffman and Sober, 2014: the effects of pitch shifts for one target syllable has decreasing effects on the pitch of other syllables depending on the distance from the target syllable), and it will be important to reveal the extent to which manipulations of timing differ from manipulations of syllable structure. The authors should also compute the level of interference in behavioral experiments akin to that computed for Figures 2K and 3I. (Or conduct more behavioral experiments to allow for such a matrix to be created).

2. The authors should compare and contrast their data with the results reported in Glaze and Troyer (2006). In particular, the authors need to address correlations in syllable timing across adjacent syllables. One finding from their detailed analysis of zebra finch song tempo was that the duration of adjacent syllables was more correlated than the duration of syllables two positions away. While there was inter-bird variation in the strength of this relationship, the overall significant relationship speaks to the relative dependence of timing across temporally proximal intervals.

Minor points:

line 238: what gFB leads to variation in timing (i.e., degree of temporal precision) that is comparable to the observed data? I ask because I'm curious to know the degree of specificity and interference of the architecture (gFB) that best approximates the precision observed in real data.

line 252: Why was feedback input periodic? How biologically plausible is this?

Line 261 and Figure 2D. Can the authors be more specific about what P is? Is it the magnitude of the perturbation or robustness to the perturbation?

For the dynamic attractors, the authors discussed how the interference in the adjacent intervals was much larger than the average interference. This is similar to findings in Glaze and Troyer (2006) in which they reported that the duration of adjacent syllables was more correlated than the duration of syllables two positions away. Please discuss.

lines 440-444. The interference observed in the synfire chain (Figure 5F) was said to be less than that observed for RNN models and comparable to that observed in birds. While it is clearly the case that the interference is lower for the synfire chain model than other models, interference observed in the synfire model appears larger than that observed in the real data. Could the authors speak to this?

lines 453-454. The fact that standard deviations changed with means is not surprising. Unless the authors justify the analysis of non-normalized standard deviations, the authors should focus their results section on how CV (normalized standard deviations) change with means.

The authors investigate the relationship between mean and variability of intervals for multiple types of fsRNNs and dynamic attractor models. But they only present the data for one type of synfire chain. I would like to see how this relationship could change with, for example, synaptic weights in the synfire chain model. You can, indeed, find linear relationships in other models (e.g., first panel of Figure 6D). In other words, the robustness of the finding was not explicitly tested for the synfire chain model.

Figure 1B. It will be useful to see more data from experiments in which the first (or another early) segment was targeted. Figure 1B&C depict the results from an experiment in which the last interval in the motif was targeted; this means that all non-targeted intervals occurred before the target interval. One possibility is all intervals AFTER the target interval might be affected in some way, and it would be useful to see some data (or depiction of data) that allow for this comparison.

Figure 1C. How many days after CAF is this data from? What happens earlier in the learning process? Is there independence throughout?

Figure 6B. The true relationship between "segment variability" and "segment duration" cannot be linear since the y-intercept should be a negative standard deviation, which is not possible. Please discuss or model using a different curve. Or remove (see above).

Figure 6C. There seems to be pseudoreplication in authors' analyses here. If this is data from different shifts across multiple birds, the analysis should have the ID of the bird as a random factor in the model. In short, they should use some sort of mixed effects model to account for the fact that they are including measurements from multiple segments per bird. (Unless I missed the mark for this analysis).

Reviewer #2 (Remarks to the Author):

This paper investigates the ability to change (i.e. learn) the timing of specific motor activities separately for selected parts of a sequence. Experiments in the song bird (Zebra Finch) are shown that reveal that the duration of different syllables of a song can be modified rather independently from each other by reinforcement learning. These results are then used as constraints on the putative neuronal networks generating the sequences of song segments are tested in rate based recurrent networks and feedforward networks of spiking neurons. While in the feedforward networks the experimental findings are reproducible, the considered recurrent networks exhibit systematic deviations from the experimental results.

The presentation is very clear and the experimental as well as the theoretical results appear well done to me. In particular, considering the degree of 'orthogonality' of the gradients w.r.t. the respective segment durations is nice and found to correspond qualitatively to the results obtained with simulated

reinforcement learning. The conclusion that this orthogonality is important is convincing, but also somewhat obvious. That is, that the synaptic connections responsible for generating the different segments need to be rather independent for realizing the observed independence of segment duration learning suggests itself. This does not, however, exclude recurrent networks at all but rather points to a sparse and thereby independent effective network structure, as the authors also somewhat concede when they discuss 'functionally feedforward networks'. The comparison of feedforward networks with spiking neurons with recurrent networks would be more convincing if the latter would also have been based on spiking neurons. Why hasn't this been done? In the paper they thereby compare apples with pears which does not appear fair to me. For instance, isn't it conceivable that for robustly generating sequences in recurrent networks of spiking neurons the emerging connectivity structure selforganizes into rather independent subnetworks that will suit better to the 'orthogonality' required for reproducing the experimental results? Including simulations with recurrent networks of spiking neurons would therefore strengthen the conclusions in the paper.

Reviewer #3 (Remarks to the Author):

Summary:

Authors induced adaptive changes to the temporal structure of birdsong (i.e. duration of song segments), then they reported that the duration of specific song segments can be modified without affecting the timing in other parts of the song. They tested what types of neural networks could explain this behavioral result, and found that randomly connected recurrent neural networks (RNN) did not generally simulate the behavioral results, whereas feedforward networks (or synfire chain model) are better suited. From those observations, they proposed general prescriptions for pattern generator networks that implement flexible motor timing.

General comments:

The present manuscript describes important and significant results and interpretations in the research field of behavioral and computational neuroscience. I comment mainly on methodological aspects of the behavioral experiments, and some logical points connecting behavioral data with the network model experiments, since I'm not an expert on the neural network study and I cannot judge relevancy on mathematical/computational details of their models.

My main concern is that their model does not take account of spontaneous (not artificial) reinforcements for maintaining songs as explained below. The critical point of the authors' argument is an independency of control in segment duration. They demonstrated that birds stretched/shrank only CAF-targeted song segments, but not non-target segments. They speculated that this target-specific change reflected the specificity in neural circuits for timing control, and used this as the most important constraint to determine the neural network model. However, it is possible that the behavioral specificity is not a consequence of the specificity in neural mechanism. Previous studies have reported that the song is recovered to its original status both in pitch and in duration after terminating the noise feedback (e.g. Ali et al 2013 Neuron 80, 494–506), suggesting that bird spontaneously reinforces itself to keep the original song. This spontaneous reinforcement could contribute to keep the duration of non-target segments even if the neural mechanism does not have segment-specific representation. In other words, the spontaneous reinforcement can cancel out the change in non-target segments that caused by the interference from CAF-associated changes in the target segment. The authors should address to this possibility since their neural network model did not cover the spontaneous reinforcement.

The authors need to explain clearly how they defined song segments. They described as "we mostly

targeted 'syllable + gap' segments" (line 606-607) but it is important to discriminate among syllable, gap, syllable+gap, and gap+syllable. The authors sometime chose gaps as the target segment (as shown in Fig. 1E), although several studies have shown that the duration of the gap has an interaction (or trade-off relationship) with the duration of surrounding syllables (e.g. Glaze and Troyer 2006 JNS 26, 991-1005; Tachibana et al 2015 JCP-A 201, 1157-1168). This contamination in song segment units could weaken the interpretation of behavioral data on the specificity in timing control. Related to this issue, it would be better if the authors could show the data about non-target segments that were adjacent to the target segment, in order to assess the "interference" effect and to show the independency of CAF-associated duration change.

I don't understand why they chose recurrent neural networks (RNN) to compare with the feedforward (synfire chain) model. Previous studies have suggested that behavioral data and neural activity in HVC can be explained by the synfire chain model (e.g. Long et al 2008 Nature 456, 189-194; Long et al 2010 Nature 468, 394-399), although the authors cited only one paper proposed RNN model for a song sequence generation (Yamashita et al 2008). It would be better for readers if they could add some explanations of how RNN is relevant for the comparison with the synfire model in the context of song sequence generation.

Specific comments:

Endash should be used to show ranges of values (e.g. 10-20 ms) instead of hyphen.

Space (non-breaking space) should be inserted between value and unit throughout the entire manuscript.

In line 627-627, I could not understand why and what they did here to remove circadian effect.

In line 630-631, it would be better if they could clarify how they defined and removed the baseline drift. Did they calculate the baseline drift as an absolute change during several days of without-CAF period? But, this value seems to vary if number of days is changed. Further, it would be helpful if they add brief explanations of the baseline in Figure 1.

In Figure 1, it would be better if a brief explanation about "alone" and "together" conditions can be added.

In Figure 6, it is hard to understand what they want to show in panel B and C. Why they used SD in panel B, but CV in panel C? What is the difference of them? What is the reason why they did not use CV in B.

We thank the reviewers for their insightful comments, and are very pleased that they find the work “elegant” (Reviewer 1), “well done” (Reviewer 2), and “important and significant... in the research field of behavioral and computational neuroscience” (Reviewer 3). In response to the reviewers’ comments, we have now significantly revised the manuscript to substantially improve the study’s scientific findings and significance. The major revisions on the experimental component include a more detailed analysis of the non-target changes as a function of position in song as well as correlations with baseline timing changes, which were asked for by two of the reviewers. New network simulations were done to explore how flexible learning is affected by 1) internally generated rewards in recurrent networks and 2) variations of architecture in synfire chains.

See below for our point-by-point responses.

Reviewer #1 (Remarks to the Author):

1. While the behavioral experiments are intriguing, they are less fleshed out than computational models. Currently, while the authors provide some data to allow comparisons between computational models and behavior, the behavioral analysis is much more superficial. For example, it is important to plot something akin to the interference (flexibility) matrices (e.g., Figure 2E) for the observed behavioral data. This will allow one to more directly compare the results from computational models with the observed data. This will also allow the reader to evaluate the degree to which the distance from the target interval could affect timing (see also comment below). Such a relationship has been demonstrated for the control of syllable structure (e.g., Hoffman and Sober, 2014: the effects of pitch shifts for one target syllable has decreasing effects on the pitch of other syllables depending on the distance from the target syllable), and it will be important to reveal the extent to which manipulations of timing differ from manipulations of syllable structure. The authors should also compute the level of interference in behavioral experiments akin to that computed for Figures 2K and 3I. (Or conduct more behavioral experiments to allow for such a matrix to be created).

We thank the reviewer for incentivizing us to explore the behavioral data further. We now plot the changes in durations relative to baseline drift for all segments as a function of distance to target (Figure 1D). None of the non-target segments changed significantly more than baseline drift (one sample t-test, $0.24 \leq p_s \leq 0.98$), thus there is no relationship between interference and non-target distance to target. Note that there are more non-targets before the target segment than after because of the design of the tCAF software: we needed to match as much of the song motif before the target using a template matching algorithm in order to reliably detect the target.

Regarding the comments about the simulation figures, note that not all of the simulation matrices can be obtained from the actual birdsong data. For example, in Figure 2E each segment is compared against all others whereas in actual birdsong learning, this is simply not feasible. For example, to reliably target the 1st segment of the song would be near impossible since there is no song template to match before its onset at the start of a song bout. Indeed we screen birds based on our ability to reliably detect song segments on-line, and it is rare to get more than 1-2 such segments per bird.

2. The authors should compare and contrast their data with the results reported in Glaze and Troyer (2006). In particular, the authors need to address correlations in syllable timing across adjacent syllables. One finding from their detailed analysis of zebra finch song tempo was that the duration of adjacent syllables was more correlated than the duration of syllables two positions away. While there was inter-bird variation in the strength of this relationship, the overall significant relationship speaks to the relative dependence of timing across temporally proximal intervals.

We thank the reviewer for bringing up this issue. The papers by Glaze and Troyer (2006, 2007, 2012, 2013) are about correlations in timing of song segments in normal undirected singing, i.e. baseline timing correlations. This is different from what we parse in our study, which deals with adaptive modifications of song using conditional auditory feedback, i.e., learning. However, we agree that the two may be related. To get at this we analyzed in greater detail how non-target changes (interference) depended on 1) the baseline timing correlation between target and non-target (Supplementary Figure 1) or 2) the extent of target changes or 3) both. In neither of these cases was there a dependence, i.e. non-target changes were not larger when the baseline timing correlation between target and non-target interval durations was higher ($r = -0.008$, $p = 0.93$). Non-target changes were also not larger when the targets were modified to a greater extent ($r = -0.14$, $p = 0.11$). It is possible that both the baseline timing correlations and target changes jointly contribute to non-target changes, i.e. non-targets that have the highest correlation with the target when the target was modified the most might exhibit most interference. Using multiple regression to test this hypothesis, we found no correlation of either factor with non-target duration changes ($R^2 = 0.02$, $p = 0.28$). Overall, our detailed analysis strongly suggests specificity in birdsong timing, i.e., modifications to one part of a sequence leaves the temporal structure of other parts unaffected regardless of any baseline timing correlations. These data have been included in the supplementary information.

Minor points:

3. line 238: what gFB leads to variation in timing (i.e., degree of temporal precision) that is comparable to the observed data? I ask because I'm curious to know the degree of specificity and interference of the architecture (gFB) that best approximates the precision observed in real data.

The noise is adjusted such that $gFB = 1$ leads to a 1% coefficient of variation of temporal variability, consistent with the 'independent variability' observed in zebra finch song (Glaze and Troyer, 2012). This value is used for Figure 6 (as indicated in the figure), where learning related changes in variability is compared to songbirds. This is now clarified in the the figure caption.

4. line 252: Why was feedback input periodic? How biologically plausible is this?

Thanks for bringing this up. It is standard procedure in feedback stabilized networks (or echo-state networks as named by (Maass et. al. 2002)) to feed the output of the network back to itself. The feedback stabilizes the network activity, which otherwise would be chaotic, and allows trainability (Sussillo & Abbott 2009, Rajan et. al. 2010). In our case the output (and hence the feedback) is a periodic

signal, which is a biologically plausible mechanism to divide time into intervals, as discussed by for example Goel and Buonomano (2014).

5. Line 261 and Figure 2D. Can the authors be more specific about what P is? Is it the magnitude of the perturbation or robustness to the perturbation?

We thank the reviewer for bringing this to our attention. P is the magnitude of the perturbation as described in the figure legend. We changed the Figure title from “Robustness to Perturbation (P)” to “Robustness to Perturbation” to avoid confusion.

6. For the dynamic attractors, the authors discussed how the interference in the adjacent intervals was much larger than the average interference. This is similar to findings in Glaze and Troyer (2006) in which they reported that the duration of adjacent syllables was more correlated than the duration of syllables two positions away. Please discuss.

As discussed above, the papers by Glaze and Troyer (2006, 2007, 2012, 2013) are about correlations in timing in normal undirected singing. We did not see any dependence of non-target changes (interference) on baseline timing correlations between targets and non-targets (new Supplementary Figure 1). This point is now mentioned in the main text.

7. lines 440-444. The interference observed in the synfire chain (Figure 5F) was said to be less than that observed for RNN models and comparable to that observed in birds. While it is clearly the case that the interference is lower for the synfire chain model than other models, interference observed in the synfire model appears larger than that observed in the real data. Could the authors speak to this?

Thanks for the careful reading. The interference in the synfire chain model is less than 1%, which is very low. We now acknowledge in the text that this is still not zero.

Also note that, in the songbird data (as replotted in Figure 1D), non-target intervals also changed. These changes were not significantly different than what would be observed due to naturally occurring drifts on baseline interval durations, therefore we cannot tell whether these changes are due to interference or not. Nevertheless, if we were to calculate an interference percentage using these changes, it would be 0.68%, close to the synfire chain model.

8. lines 453-454. The fact that standard deviations changed with means is not surprising. Unless the authors justify the analysis of non-normalized standard deviations, the authors should focus their results section on how CV (normalized standard deviations) change with means.

While it might be true that the standard deviations changing with means is not surprising at the behavioral level (something found in many motor timing tasks), it is clear that recurrent networks, most of the time, do not reproduce this ubiquitous phenomenon, which is our major point in Figure 6. A lot of

the time standard deviations decrease with increasing means. Our initial presentation of this point was confusing, therefore we replotted the data in the figure and updated the discussion in the text.

9. The authors investigate the relationship between mean and variability of intervals for multiple types of fsRNNs and dynamic attractor models. But they only present the data for one type of synfire chain. I would like to see how this relationship could change with, for example, synaptic weights in the synfire chain model. You can, indeed, find linear relationships in other models (e.g., first panel of Figure 6D). In other words, the robustness of the finding was not explicitly tested for the synfire chain model.

We thank the reviewer for bringing this point up. We added two variants of the synfire chain model: 1) Random weights: all-to-all connectivity from one layer to the next but the weights are chosen randomly, and 2) Random connectivity: a neuron makes a synapse to a neuron in the layer with probability 0.8. In both cases we see that the interference matrix is diagonal, interference is ~1%, and variability is a monotonic function of the interval duration. We have now added this analysis to the paper, and the resulting plots can be found in Supplementary Figure 4.

10. Figure 1B. It will be useful to see more data from experiments in which the first (or another early) segment was targeted. Figure 1B&C depict the results from an experiment in which the last interval in the motif was targeted; this means that all non-targeted intervals occurred before the target interval. One possibility is all intervals AFTER the target interval might be affected in some way, and it would be useful to see some data (or depiction of data) that allow for this comparison.

In Figure 1B & C, the second from last syllable+gap segment was targeted. There is still a syllable after the target (the noise feedback in 1B “hides” the last syllable after the target). We now plot the changes in durations relative to baseline drift for all segments as a function of distance to target (Figure 1D). None of the non-target segments changed significantly more than baseline drift (one sample t-test, $0.24 \leq p \leq 0.98$), thus there is no discernable relationship between interference and non-target distance to target.

11. Figure 1C. How many days after CAF is this data from? What happens earlier in the learning process? Is there independence throughout?

The CAF experiments were done after 6-10 days of CAF. We examined the data in the first few days of CAF and also did not find any interference. This is now discussed.

12. Figure 6B. The true relationship between “segment variability” and “segment duration” cannot be linear since the y-intercept should be a negative standard deviation, which is not possible. Please discuss or model using a different curve. Or remove (see above).

We thank the reviewer for this comment. Instead of Pearson correlation coefficient, which measures linear correlation between two variables, we now report Kendall’s tau which measures a general monotonic relationship.

13. Figure 6C. There seems to be pseudoreplication in authors' analyses here. If this is data from different shifts across multiple birds, the analysis should have the ID of the bird as a random factor in the model. In short, they should use some sort of mixed effects model to account for the fact that they are including measurements from multiple segments per bird. (Unless I missed the mark for this analysis).

There is no pseudoreplication in Figure 6C. Each animal contributes a single datapoint – the average change in target duration over the whole CAF. We have clarified this in the caption. This contrasts with Figure 6B, which is a figure for an example animal, thus has the individual day's duration and variability plotted.

Reviewer #2 (Remarks to the Author):

1. The presentation is very clear and the experimental as well as the theoretical results appear well done to me. In particular, considering the degree of 'orthogonality' of the gradients w.r.t. the respective segment durations is nice and found to correspond qualitatively to the results obtained with simulated reinforcement learning. The conclusion that this orthogonality is important is convincing, but also somewhat obvious. That is, that the synaptic connections responsible for generating the different segments need to be rather independent for realizing the observed independence of segment duration learning suggests itself. This does not, however, exclude recurrent networks at all but rather points to a sparse and thereby independent effective network structure, as the authors also somewhat concede when they discuss 'functionally feedforward networks'. The comparison of feedforward networks with spiking neurons with recurrent networks would be more convincing if the latter would also have been based on spiking neurons. Why hasn't this been done? In the paper they thereby compare apples with pears which does not appear fair to me. For instance, isn't it conceivable that for robustly generating sequences in recurrent networks of spiking neurons the emerging connectivity structure selforganizes into rather independent subnetworks that will suit better to the 'orthogonality' required for reproducing the experimental results? Including simulations with recurrent networks of spiking neurons would therefore strengthen the conclusions in the paper.

The reviewer is pointing to important issues that we had failed to clarify in the main text. We want to present our answer under separate titles.

A. Are we comparing apples to pears?

We respectfully disagree with this contention. A spiking network can be approximated by a rate network and capture its fundamental dynamics (Abbott et. al. 2016, see quote below). This is the reason rate networks are used as models in neuroscience. This notion can be formalized at least in two ways:

- 1) In the simplest and most direct scenario, a rate model can be thought to represent the average firing rate of many spiking neurons. Therefore, our trained recurrent networks with rate neurons can be mapped to a spiking recurrent network where each rate neuron is replaced with

hundreds to thousands spiking units. In this sense, a recurrent rate network is merely a coarse-grained version of its spiking counterparts.

- 2) A more efficient mapping from a rate network to a spiking network has been described recently in (DePasquale *et. al.*, 2016). Given a trained rate network that solves a task, DePasquale *et. al.* uses it to find the inputs to the spiking neurons that should be trained. Such network ends up mimicking the dynamics of the rate network they are built on, with much fewer neurons than the simple scenario described above. Quoting (Abbott *et. al.* 2016).: “Continuous-variable (rate) networks not only play a key role in the construction of these spiking networks but also describe the fundamental dynamics signals by which the spiking networks operate. This makes them well suited for describing how neural circuits operate, not mechanistically (spiking networks are closer to this) but at a basic functional level”.

Both these approaches show that a trained recurrent rate network captures the fundamental dynamics of the corresponding spiking network, but obviously not all the mechanistic details. As long as the mechanistic details are not relevant for the argument, the conclusions made about rate networks should carry over to their spiking counterparts. This is the case for the trained rate networks in our paper.

- 1) In fsRNNs, the mechanism behind the observed interference phenomena is resetting of network state due to feedback.
- 2) In dynamic attractors, interference is due to the collective organization of the network to a dynamic attractor.

In contrast, for a synfire chain the mechanistic details matter. Synchronization of neurons in a node, and the propagation of spike volleys without dispersing or diminishing are all contingent of neurons generating spikes. A chain of rate neurons would not be able to capture these dynamical properties. Therefore, simulating a synfire chain with spiking neurons is necessary.

B. If rate networks are the “same” as their spiking counterparts, why haven’t we trained and simulated spiking networks directly?

This is mainly a computational resource issue, simulating spiking networks is computationally much costlier. Therefore, we chose to simulate spiking networks only when spiking dynamics is relevant to the point we are making, as in synfire chains.

C. “isn't it conceivable that for robustly generating sequences in recurrent networks of spiking neurons the emerging connectivity structure self-organizes into rather independent subnetworks that will suit better to the 'orthogonality' required for reproducing the experimental results?”

The answer to this question depends on the method of training, as can be seen in our rate network examples, where different ways of training (feedback stabilization vs. dynamic attractor) leads to different interference patterns. The state-of-the art spiking network training method (DePasquale *et. al.*, 2016, Abbott *et. al.* 2016) transfers the dynamics from a rate network to a spiking network and hence

spiking networks constructed this way would mimic the interference patterns of the underlying rate networks.

On the other hand, it is plausible that a yet to be invented spiking network training method will find an orthogonal, non-interfering solution, which is not a feedforward network. In our discussion, we speculated one example of such a solution: a functionally feedforward architecture with a dynamics that generalizes synfire chains.

In the light of the above points, we think that spiking simulations are not necessary to support our conclusions in this paper. And though methods for training spiking recurrent network are lacking and would be a valuable addition to the computational neuroscientist's toolbox, we believe that that task is well beyond the scope of our paper, and, more importantly, would change its focus and detract from its main message.

Reviewer #3 (Remarks to the Author):

General comments:

The present manuscript describes important and significant results and interpretations in the research field of behavioral and computational neuroscience. I comment mainly on methodological aspects of the behavioral experiments, and some logical points connecting behavioral data with the network model experiments, since I'm not an expert on the neural network study and I cannot judge relevancy on mathematical/computational details of their models.

1. My main concern is that their model does not take account of spontaneous (not artificial) reinforcements for maintaining songs as explained below. The critical point of the authors' argument is an independency of control in segment duration. They demonstrated that birds stretched/shrank only CAF-targeted song segments, but not non-target segments. They speculated that this target-specific change reflected the specificity in neural circuits for timing control, and used this as the most important constraint to determine the neural network model. However, it is possible that the behavioral specificity is not a consequence of the specificity in neural mechanism. Previous studies have reported that the song is recovered to its original status both in pitch and in duration after terminating the noise feedback (e.g. Ali et al 2013 Neuron 80, 494–506), suggesting that bird spontaneously reinforces itself to keep the original song. This spontaneous reinforcement could contribute to keep the duration of non-target segments even if the neural mechanism does not have segment-specific representation. In other words, the spontaneous reinforcement can cancel out the change in non-target segments that caused by the interference from CAF-associated changes in the target segment. The authors should address to this possibility since their neural network model did not cover the spontaneous reinforcement.

The reviewer raises an important point. The analysis presented in the paper applies to internally generated (or 'spontaneous') reinforcements as well as CAF-associated ones. Specifically, Eq. 10 holds regardless of the nature of the reinforcement.

To be more concrete, we now present new recurrent network simulations in Supplementary Figures 2F, 2G, 3E and 3F. We introduced internally generated reinforcement signals for all intervals, as a proxy for the learning process driven by deviations from an internally held template. We tuned the strength of this reinforcement such that when tCAF is turned off, the target interval returned to baseline. As observed in (Ali et al 2013 *Neuron* 80, 494–506) we tuned the parameters such that the return to baseline was ~3 times slower than tCAF. We found that such internal reinforcement do not necessarily decrease interference. Indeed, in many cases it increased it. This can be understood through Eq. 10. Even though an interval reinforcer may prefer to keep its target interval at its baseline value, interference from other internal reinforcers may override it. This is now discussed in the text.

*The authors need to explain clearly how they defined song segments. They described as “we mostly targeted ‘syllable + gap’ segments” (line 606-607) but it is important to discriminate among syllable, gap, syllable+gap, and gap+syllable. The authors sometime chose gaps as the target segment (as shown in Fig. 1E), although several studies have shown that the duration of the gap has an interaction (or trade-off relationship) with the duration of surrounding syllables (e.g. Glaze and Troyer 2006 *JNS* 26, 991-1005; Tachibana et al 2015 *JCP-A* 201, 1157-1168). This contamination in song segment units could weaken the interpretation of behavioral data on the specificity in timing control. Related to this issue, it would be better if the authors could show the data about non-target segments that were adjacent to the target segment, in order to assess the “interference” effect and to show the independency of CAF-associated duration change.*

Reviewer 1 brought up similar issues regarding relative position of non-targets and we repeat our response here. We now plot the changes in durations relative to baseline drift for all segments as a function of distance to target (Figure 1D). None of the non-target segments changed significantly more than baseline drift (one sample t-test, $0.24 \leq p \leq 0.98$), thus there is no relationship between interference and non-target distance to target.

*I don't understand why they chose recurrent neural networks (RNN) to compare with the feedforward (synfire chain) model. Previous studies have suggested that behavioral data and neural activity in HVC can be explained by the synfire chain model (e.g. Long et al 2008 *Nature* 456, 189-194; Long et al 2010 *Nature* 468, 394-399), although the authors cited only one paper proposed RNN model for a song sequence generation (Yamashita et al 2008). It would be better for readers if they could add some explanations of how RNN is relevant for the comparison with the synfire model in the context of song sequence generation.*

We aimed this study to be more general than song sequence generation. RNNs have been proposed as sequence generators in many papers, some of which are cited in our introduction. For the particular case of the songbird sequence generation, the synfire chain model is the most widely accepted model. But while experimental data is consistent with the synfire chain model, much of this experimental data is not inconsistent with RNN. We believe our analysis provides crucial additional evidence in favor of the synfire chain model. This point is now also made more explicit in the text, at the last sentence of the introduction.

Specific comments:

Endash should be used to show ranges of values (e.g. 10-20 ms) instead of hyphen.

Done. Thanks.

Space (non-breaking space) should be inserted between value and unit throughout the entire manuscript.

Done. Thanks.

In line 627-627, I could not understand why and what they did here to remove circadian effect.

We simply analyzed data from the same time of the day (e.g., morning's song with the next morning's song) to rule out effects of time of day on segment durations and variability. This is explained in Methods.

In line 630-631, it would be better if they could clarify how they defined and removed the baseline drift. Did they calculate the baseline drift as an absolute change during several days of without-CAF period? But, this value seems to vary if number of days is changed. Further, it would be helpful if they add brief explanations of the baseline in Figure 1.

Baseline was calculated over the same number of days as CAF (6-10 days) before animals underwent CAF. Note that for all changes in duration (CAF or baseline), they are divided by number of days (ms/day) and not absolute total change. These details of the analysis have now been clarified in both Methods and Figure 1 legend.

In Figure 1, it would be better if a brief explanation about "alone" and "together" conditions can be added.

Thanks for the comment. The caption for Figure 1G already states what alone and together means. The methods section further explains the experiment.

In Figure 6, it is hard to understand what they want to show in panel B and C. Why they used SD in panel B, but CV in panel C? What is the difference of them? What is the reason why they did not use CV in B.

We thank the reviewer for bringing this up. The point we were trying to make is that when segment intervals are shortened, variability in their durations decrease, whereas when segments are lengthened their variability increase. This point can be made by standard deviation alone, therefore we now only plot standard deviation in both panels. Note that Figure 6B plots an example animal showing the

monotonic relationship between variability and segment duration across days during CAF. Figure 6C is the overall data from 18 animals showing the relationship holds.

Reviewers' comments:

Reviewer #1 (Remarks to the Author):

In general, the authors did a good job addressing my concerns. The manuscript is clearer, and the authors include explanations and justifications that make the manuscript more accessible to a larger audience. However, I ask that the authors include more raw data and conduct a few additional analyses before publications.

Figure 2D is an improvement over previous depictions of the behavioral data. This data seems overall compelling, but it is important for the readers to get a sense of what the non-normalized data look like. While I understand the relevance of subtracting baseline drift, it leaves the reader wondering what the data look like before the subtraction. Given how compelling the data appear, I suspect it will look fine, so the authors should provide this data in a supplementary figure. The authors already present some non-normalized data in the manuscript (e.g. Supplementary Figure 1; albeit not with respect to position relative to the target interval) so this should be reasonable to depict. Another impetus for doing this is that the data they present for the models are not non-normalized data.

The authors acknowledge the contributions of Glaze and Troyer and attempt to identify how their experiment and interpretations differ from their previous findings. The authors include a number of analyses in the revision that attempt to address this, but I believe their approach should be improved. The rationale, as I see it, is as such. One would predict that, the greater the correlation in baseline timing between a target interval and a non-target interval in song, the more correlated the experimental change in the target interval would be to "drift" in the non-target interval. In other words, non-target intervals that are strongly positively correlated with the target interval should show changes in timing that are similar in sign and correlated in magnitude to extent of experimental lengthening or shortening of the target interval. In contrast, non-target intervals that are strongly negatively correlated with the target interval should show changes in timing that are opposite in sign but correlated in magnitude following experimental lengthening or shortening of the target interval, and non-target intervals that are not correlated in baseline timing with the target interval should show no changes in timing or changes in timing that are unrelated to changes in the target interval. However, the authors don't quite conduct this analysis, even with their multiple regression (at least as far as I can tell). In total, three aspects need to be taken into consideration: the baseline correlation between the target and non-target interval, the degree of change in the target interval, and the degree of change in the non-target interval. Supplementary Figure 1a analyzes the relationship between the "baseline correlation between target and non-target interval durations" and the "non-target duration change". However, what is unclear from these data is how much the corresponding target interval changed for these data. On the other hand, Supplementary Figure 1b analyzes the relationship between how much the corresponding target interval changed and the "non-target duration change". However, what is unclear from this data are the baseline correlation between target and non-target interval durations for these corresponding points. The multiple regression is a step in the right direction, but the interaction between the two explanatory terms should be included in the model. In other words, how the the magnitude and direction of non-target interval changes relates to the magnitude and direction of target interval changes should vary depending on the sign and strength of the baseline correlation between these factors. Consequently, I strongly encourage the authors to re-examine the relationship with this in mind.

Minor points:

The specificity of tempo changes is reminiscent of specificity reported following acute feedback perturbations in a related species, the Bengalese finch (Sakata and Brainard, 2006). The authors should consider citing this paper because it supports the notion of specificity in vocal control and

speaks to the generality of this finding.

Figure 1F: Shouldn't there be error bars for each of these points? The figure legend reads as if the figure is supposed to summarize population data. In fact, it says the panel plots mean+SEM but I can't see error bars.

Figure 1G: There is pseudoreplication in this analysis (12 segments measured in 6 birds), so the authors should use a type of repeated-measures model (e.g., mixed effects model). Given the trends, it probably shouldn't affect the results, but the authors should analyze their data in a way that is consistent with the structure of the data.

line 316: "?" should be "."

Figure 6F: The magnitude of interval variability in the model seems to be different than what you would predict from the behavioral data. Actually, it's a little hard to tell since the relationship between interval duration and variability is non-linear (see comment from original submission). But the discrepancy between this panel and 6B is noticeable. Please address.

line 540-544: I urge the authors to clarify how a synfire chain can be implemented in a distributed manner across brain areas. This statement can be interpreted as if downstream brain areas (further down in the synfire chain) could regulate later parts of the song, while upstream areas encode earlier parts of the song. There is no support for this idea, as far as I know.

line 1235: Figure legend for Supp. Figure 2B: Should be "absolute change in average learning" (as is noted in the figure).

Supp. Figure 2B: It seems like the sign of the change is important. For example, if experimentally decreasing the 8th interval interferes with increase in the 3rd interval, you should see a lower learning rate for the 3rd interval (i.e., negative change). Is this the case for this model? Please indicate. This is a general comment regarding the use of absolute changes (e.g., also Supp. Figure 4H), since directionality of interference is important.

The r value in Supp. Figure 2C is a Pearson's correlation? Seems like the authors should use a non-parametric since the relationship is non-linear. The same can be said about a number of panels (e.g., in Supp. Fig. 1E). Please note in the legend what correlation function is used.

Reviewer #2 (Remarks to the Author):

The discussion now nicely clarifies that RNNs which functionally realise synfire chains could potentially also explain the data. With this and by taking the other referees comments into account I now consider this research very clear and well done and thereby suitable for publication.

Reviewer #3 (Remarks to the Author):

The manuscript has improved very well. I have nothing more to point out.

We are very encouraged that all three reviewers found the manuscript substantially improved. Below we detail responses to Reviewer 1.

Reviewer #1 (Remarks to the Author):

In general, the authors did a good job addressing my concerns. The manuscript is clearer, and the authors include explanations and justifications that make the manuscript more accessible to a larger audience. However, I ask that the authors include more raw data and conduct a few additional analyses before publications.

Figure 2D is an improvement over previous depictions of the behavioral data. This data seems overall compelling, but it is important for the readers to get a sense of what the non-normalized data look like. While I understand the relevance of subtracting baseline drift, it leaves the reader wondering what the data look like before the subtraction. Given how compelling the data appear, I suspect it will look fine, so the authors should provide this data in a supplementary figure. The authors already present some non-normalized data in the manuscript (e.g. Supplementary Figure 1; albeit not with respect to position relative to the target interval) so this should be reasonable to depict. Another impetus for doing this is that the data they present for the models are not non-normalized data.

We agree with the reviewer that presenting data without baseline subtraction changes would be informative for readers. We now include the data in Supplementary Figure 1A. Given this new panel in Supplementary Figure 1, the references to each panel in the manuscript have been changed accordingly. Note also the minor cosmetic change to the dots in panels 2 and 3 (from red to the color scheme used in Figure 1D).

The authors acknowledge the contributions of Glaze and Troyer and attempt to identify how their experiment and interpretations differ from their previous findings. The authors include a number of analyses in the revision that attempt to address this, but I believe their approach should be improved. The rationale, as I see it, is as such. One would predict that, the greater the correlation in baseline timing between a target interval and a non-target interval in song, the more correlated the experimental change in the target interval would be to “drift” in the non-target interval. In other words, non-target intervals that are strongly positively correlated with the target interval should show changes in timing that are similar in sign and correlated in magnitude to extent of experimental lengthening or shortening of the target interval. In contrast, non-target intervals that are strongly negatively correlated with the target interval should show changes in timing that are opposite in sign but correlated in magnitude following experimental lengthening or shortening of the target interval, and non-target intervals that are not correlated in baseline timing with the target interval should show no changes in timing or changes in timing that are unrelated to changes in the target interval. However, the authors don’t quite conduct this analysis, even with their multiple regression (at least as far as I can tell). In total, three aspects need to be taken into consideration: the baseline correlation between the target and non-target interval, the degree of change in the target interval, and the degree of change in the non-target interval. Supplementary Figure 1a analyzes the relationship between the “baseline correlation between target and non-target interval durations” and the “non-target duration change”. However, what is unclear from these data is how much

the corresponding target interval changed for these data. On the other hand, Supplementary Figure 1b analyzes the relationship between how much the corresponding target interval changed and the “non-target duration change”. However, what is unclear from this data are the baseline correlation between target and non-target interval durations for these corresponding points. The multiple regression is a step in the right direction, but the interaction between the two explanatory terms should be included in the model. In other words, how the the magnitude and direction of non-target interval changes relates to the magnitude and direction of target interval changes should vary depending on the sign and strength of the baseline correlation between these factors. Consequently, I strongly encourage the authors to re-examine the relationship with this in mind.

To address whether the baseline correlation between the target and a non-target interval (independent variable 1) and the degree of change in the target interval (independent variable 2) can account for the change seen in the non-target interval (dependent variable), we re-ran the multiple regression with an added interaction term between the two independent variables as suggested by the reviewer. Because these two explanatory variables are continuous, we followed standard procedure to simply multiply the two variables to produce an interaction term as another independent variable. Implementing this in Matlab using the *regress* function, the multiple regression was not significant ($R^2 = 0.052$, $p = 0.10$). We report this result in the caption of Supplementary Figure 1.

Minor points:

The specificity of tempo changes is reminiscent of specificity reported following acute feedback perturbations in a related species, the Bengalese finch (Sakata and Brainard, 2006). The authors should consider citing this paper because it supports the notion of specificity in vocal control and speaks to the generality of this finding.

We thank the reviewer for bringing this paper to our attention. While the paper and its follow-up (Sakata and Brainard, 2008) did not quantify how specific the tempo changes were to the target syllable, the finding is still relevant to our study. We now cite and discuss both papers in the Discussion section.

Figure 1F: Shouldn't there be error bars for each of these points? The figure legend reads as if the figure is supposed to summarize population data. In fact, it says the panel plots mean+SEM but I can't see error bars.

Figure 1F is from an example animal to illustrate the learning curve while Figure 1G is the quantification per day across multiple birds. There are error bars in Figure 1 but they are barely visible. We have made the caption for Figure 1F and 1G clearer (lines 1102-1108).

Figure 1G: There is pseudoreplication in this analysis (12 segments measured in 6 birds), so the authors should use a type of repeated-measures model (e.g., mixed effects model). Given the trends, it probably shouldn't affect the results, but the authors should analyze their data in a way that is consistent with the

structure of the data.

We apologize for not being clearer about our statistical tests. There is no pseudoreplication because each bird contributed just one datapoint for each experimental condition - alone and w/other - despite the fact that there are 12 segments being targeted in 6 birds (2 segments per bird). The analysis was done as follows. For each bird, we averaged the change/day for the 2 segments in the alone condition and did the same for the w/other condition. This produced a vector of 6 numbers (one for each bird) for each of the two conditions. Because the same birds were used in both conditions, we then did a paired samples t-test between the two conditions, yielding a p-value of 0.54 as reported in the main text. In other words, the degree of freedom for the t-test is 5 (6 minus 1) and not 11 (12- 1). This two-step process involving averaging is a commonly accepted practice to prevent pseudoreplication without using more complex models. We have made our calculation clearer in the Methods (lines 673-677) and refer to them in the caption of Figure 1G.

line 316: "?" should be "."

Changed.

Figure 6F: The magnitude of interval variability in the model seems to be different than what you would predict from the behavioral data. Actually, it's a little hard to tell since the relationship between interval duration and variability is non-linear (see comment from original submission). But the discrepancy between this panel and 6B is noticeable. Please address.

We thank the reviewer for bringing this issue up.

Data from both our lab and others (e.g., Glaze & Troyer 2012, Glaze & Troyer, 2013, Andalman et al., 2011) have found that the coefficient of variation (CV, standard deviation divided by mean) of song timing in young adult zebra finches (90 to 180 dph) is ~1-10% (with gaps being more variable). Our example data in 6B falls right within this range (~5%).

One should also note that variability in song tempo is comprised of temporal variability in the pattern generator circuits and 'peripheral jitter', i.e. temporal variability in downstream areas and motor organs (Glaze & Troyer, 2012). Our model is that of a pattern generator and hence reproduces only the first of these components. Jitter is estimated to be about 2ms on average by Glaze & Troyer (2012), independent of interval duration. If we add that to 0.5ms variability observed in our model for a 50ms interval, we get $2.06ms = \sqrt{(0.5ms)^2 + (2ms)^2}$, which correspond to a CV of ~4%. Therefore, we believe that the model variability is in the ballpark with observed pattern generator variability in HVC.

We explain this in a new paragraph starting on line 732.

line 540-544: I urge the authors to clarify how a synfire chain can be implemented in a distributed manner across brain areas. This statement can be interpreted as if downstream brain areas (further down in the synfire chain) could regulate later parts of the song, while upstream areas encode earlier parts of the song.

There is no support for this idea, as far as I know.

This is an important point which attracted attention in literature recently. In the paper, we refer to Hamaguchi, Tanaka & Mooney (ref 50) who argue for a distributed implementation of the chain that regulates song timing, involving HVC, RA, VRG (brainstem), and UVA. In their model, activity propagates repeatedly across these four areas in a circular manner repeatedly to regulate timing over the whole song motif. So, one can interpret the nodes of the synfire chains as being in different brain areas and activity just circularly repeats.

line 1235: Figure legend for Supp. Figure 2B: Should be “absolute change in average learning” (as is noted in the figure).

We believe the reviewer is referring to Supp. Fig. 3B. The legend is corrected.

Supp. Figure 2B: It seems like the sign of the change is important. For example, if experimentally decreasing the 8th interval interferes with increase in the 3rd interval, you should see a lower learning rate for the 3rd interval (i.e., negative change). Is this the case for this model? Please indicate. This is a general comment regarding the use of absolute changes (e.g., also Supp. Figure 4H), since directionality of interference is important.

We believe the reviewer is referring to Supp. Fig. 3B. Indeed, the directionality can switch the sign of learning rate change, and precisely for this reason we use absolute changes in Supp. Fig. 3B and Supp. Fig. 4H. These figures show data averaged across all simulations. Otherwise, negative and positive changes might lead to a cancellation on average and may lead to an impression that there is little no interference, when the truth is the opposite. In this paper, we view being flexible as having no interference, which means changes in both directions are equally undesirable.

The r value in Supp. Figure 2C is a Pearson’s correlation? Seems like the authors should use a non-parametric since the relationship is non-linear. The same can be said about a number of panels (e.g., in Supp. Fig. 1E). Please note in the legend what correlation function is used.

We do not have a correlation reported in Supp. Fig. 2C, perhaps the reviewer meant Supp. Fig. 3C? And Supp. Fig. 2E by Supp. Fig. 1E?

In Supp. Fig.s 3C and 2E, we used Pearson’s correlation because we are comparing theory predictions to simulation results. The theory is a linear approximation; therefore, Pearson’s correlation serves to quantify how well the linear approximation performs.

We now state that in the captions for Fig. 2I, Supp. Fig 1A and B, Supp. Fig 2E and Supp. Fig 3C that we are referring to Pearson’s correlation coefficient. Throughout the whole paper, we consistently use “r” to denote Pearson correlation coefficient and “tau” to denote Kendall’s tau coefficient.

REVIEWERS' COMMENTS:

Reviewer #1 (Remarks to the Author):

The authors have addressed my concerns. I believe the manuscript is a valuable contribution to our understanding of vocal control and flexibility.